# Linearization Turns Neural Operators into Function-Valued Gaussian Processes

**Emilia Magnani** [* 1]   **Marvin Pförtner** [* 1]   **Tobias Weber** [* 1]   **Philipp Hennig** [1]

## Abstract

Neural operators generalize neural networks to learn mappings between function spaces from data. They are commonly used to learn solution operators of parametric partial differential equations (PDEs) or propagators of time-dependent PDEs. However, to make them useful in high-stakes simulation scenarios, their inherent predictive error must be quantified reliably. We introduce LUNO, a novel framework for approximate Bayesian uncertainty quantification in trained neural operators. Our approach leverages model linearization to push (Gaussian) weight-space uncertainty forward to the neural operator's predictions. We show that this can be interpreted as a probabilistic version of the concept of currying from functional programming, yielding a function-valued (Gaussian) random process belief. Our framework provides a practical yet theoretically sound way to apply existing Bayesian deep learning methods such as the linearized Laplace approximation to neural operators. Just as the underlying neural operator, our approach is resolution-agnostic by design. The method adds minimal prediction overhead, can be applied post-hoc without retraining the network, and scales to large models and datasets. We evaluate these aspects in a case study on Fourier neural operators.

## 1. Introduction

Scientific computing increasingly demands efficient representations of complex non-linear maps between functions. Examples include solution operators of parametric partial differential equations (PDEs) or the propagators of time-dependent PDEs. Operator learning is an approach to this problem that generalizes regression algorithms from finite-dimensional to infinite-dimensional function-space input-output pairs (Boullé & Townsend, 2023). Neural operators, including Fourier neural operators, have emerged as a powerful class of models for operator learning, particularly for the solution operators of PDEs (Kovachki et al., 2023). They have been applied successfully across domains including weather forecasting (Pathak et al., 2022; Bonev et al., 2023), fluid dynamics (Grady et al., 2022; Renn et al., 2023; Li et al., 2022), and automotive aerodynamics (Li et al., 2023b). Instead of learning to solve a specific PDE, these models learn the operator that maps a functional parameter of the PDE (such as initial values, boundary conditions, force fields, or material parameters) to the corresponding solution. This approach *amortizes* computational cost by learning to solve entire families of PDEs.

Although neural operators have demonstrated strong predictive capabilities, they are unable to quantify the inherent uncertainty in their predictions. Predictive uncertainty quantification is indispensable for many downstream tasks, such as decision-making in safety-critical scenarios. For example, a neural operator trained on past climate data should increase predictive uncertainty under distribution shifts due to climate change, reflecting potential losses in accuracy. Furthermore, uncertainty quantification is also useful for improving neural operator training via active learning strategies (Musekamp et al., 2025), potentially reducing the cost of generating computationally expensive numerical simulations as training data.

Extensive previous work has shown that the structured uncertainty provided by Gaussian process (GP) models is particularly suitable for such downstream tasks, including closed-form acquisition functions for active learning and Bayesian optimization for optimal selection of future queries or experiments (Garnett, 2023); enabling online model adaptation to continuously update with new data while preserving consistency with prior knowledge (Sliwa et al., 2024); facilitating sensitivity analysis through the GP's kernel structure revealing system responses to parameter changes; and seamlessly integrating with probabilistic numerical computation (Hennig et al., 2022; Pförtner et al., 2022) to quantify, marginalize, and propagate computational uncertainty.

Motivated by the capabilities of GPs, we propose LUNO, a practical yet theoretically sound framework that provides *linearized predictive uncertainty in neural operators*. LUNO

---

[*]Equal contribution [1]Tübingen AI Center, University of Tübingen, Tübingen, Germany. Correspondence to: Emilia Magnani <emilia.magnani@uni-tuebingen.de>.

*Proceedings of the 42nd International Conference on Machine Learning*, Vancouver, Canada. PMLR 267, 2025. Copyright 2025 by the author(s).

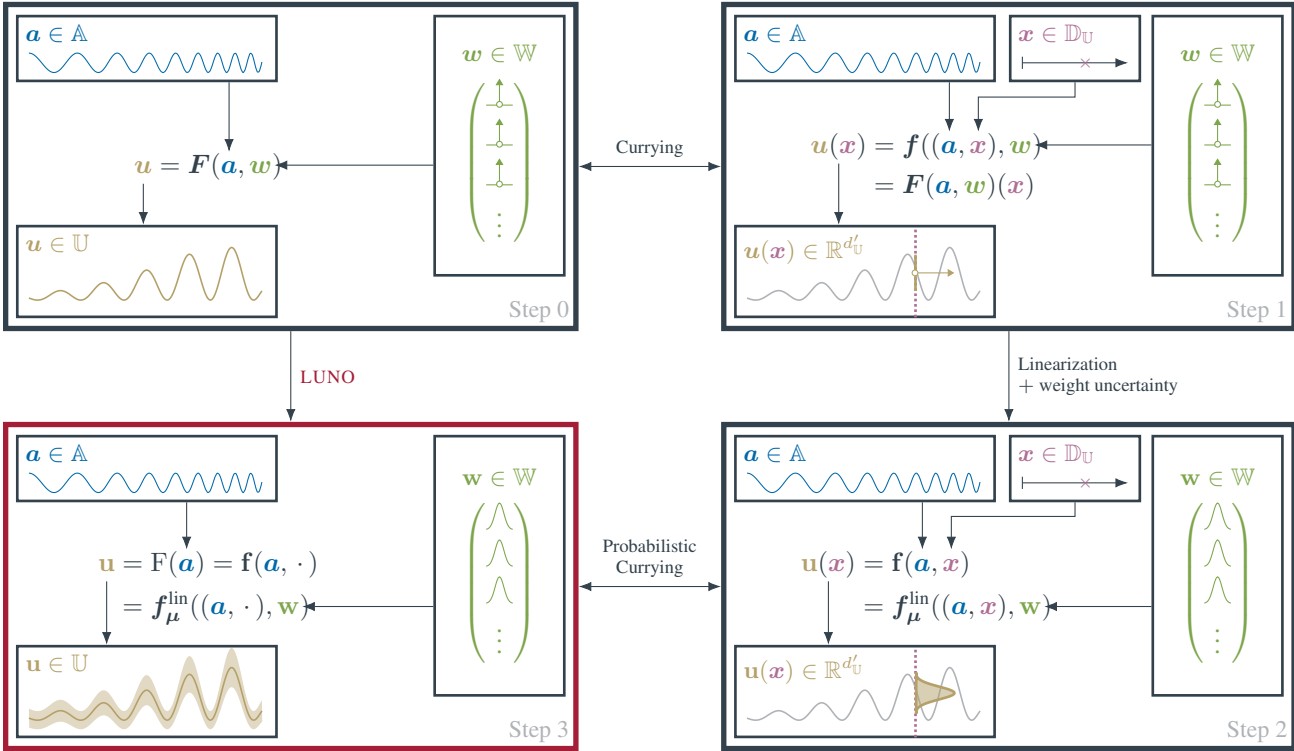

Figure 1: Illustration of the steps involved in LUNO. A trained neural operator $\boldsymbol{F}$ (**top left**) is converted into an equivalent neural network $\boldsymbol{f}$ with outputs in $\mathbb{R}^{d'_{\mathbb{U}}}$ using (reverse) currying (**top right**). Linearizing $\boldsymbol{f}$ around the mean of the Gaussian weight belief results in a Gaussian process posterior $\mathbf{f}$ quantifying the uncertainty about the function learned by $\boldsymbol{f}$ (**bottom right**). Finally, probabilistic currying transforms $\mathbf{f}$ into a function-valued Gaussian process posterior $\mathbf{F}$ over the operator learned by the neural operator $\boldsymbol{F}$ (**bottom left**).

quantifies uncertainty over the mapping learned by the neural operator via a Gaussian process with values in a separable Banach space of functions—a higher-order generalization of GPs that, when evaluated, returns a Gaussian random function rather than a finite-dimensional Gaussian random variable, as in standard GPs. This *function*-valued GP is constructed through model linearization from a Gaussian distribution quantifying uncertainty in the neural operator's (finite-dimensional) weight space. We show that LUNO can be interpreted as a probabilistic generalization of the concept of *currying* in functional programming. This connection makes LUNO compatible with established methods for quantifying weight-space uncertainty in deep neural networks, including the Laplace approximation (Ritter et al., 2018; Daxberger et al., 2021a; Kristiadi et al., 2020; Papamarkou et al., 2024), SWAG (Maddox et al., 2019), or mean-field variational inference (Blundell et al., 2015). LUNO is practical, introduces minimal computational overhead, and can be applied post-hoc, without requiring to retrain the neural operator. It scales to large models and datasets and, like neural operators, is inherently resolution-agnostic. We demonstrate the capabilities of the framework in a case

study on Fourier neural operators.

LUNO is designed to be compatible with arbitrary (non-Gaussian) weight-space beliefs (see e.g. Appendix A.4). Nevertheless, we focus our exposition and experiments on Gaussian weight-space uncertainty, as, in the future, we aim to explore the use of the resulting function-valued Gaussian process in downstream tasks such as the ones outlined above.

In Section 2, we review the fundamentals of neural operators and (multi-output) Gaussian processes. Section 3 presents our main contribution. We first develop Gaussian processes that take values in (infinite-dimensional) Banach spaces of functions, along with the notion of *probabilistic currying*, which formalizes their equivalence to (multi-output) Gaussian processes. Using probabilistic currying, we construct function-valued Gaussian processes from neural operators with Gaussian weight beliefs. In Section 4, we discuss prior work on operator learning and related uncertainty quantification. Finally, we demonstrate in Section 5 the effectiveness of LUNO in common PDE learning settings.

## 2. Background

We first review neural operators, with emphasis on Fourier neural operators, which serve as the primary case study for our analysis. Then, we provide an overview of multi-output Gaussian processes.

### 2.1. Neural Operators

*Neural operators* (NOs) (Kovachki et al., 2023) are neural network architectures that map between (infinite-dimensional) Banach spaces of functions. A neural operator is a function $\boldsymbol{F} \colon \mathbb{A} \times \mathbb{W} \to \mathbb{U}$, where

- $\mathbb{A}$ is a (separable) Banach space of functions $\boldsymbol{a} \colon \mathbb{D}_{\mathbb{A}} \to \mathbb{R}^{d'_{\mathbb{A}}}$ with domain $\mathbb{D}_{\mathbb{A}} \subset \mathbb{R}^{d_{\mathbb{A}}}$,

- $\mathbb{U}$ is a (separable) Banach space of functions $\boldsymbol{u} \colon \mathbb{D}_{\mathbb{U}} \to \mathbb{R}^{d'_{\mathbb{U}}}$ with domain $\mathbb{D}_{\mathbb{U}} \subset \mathbb{R}^{d_{\mathbb{U}}}$,

- $\mathbb{W}$ is a set of parameters (typically $\mathbb{W} \subset \mathbb{R}^p$ or $\mathbb{W} \subset \mathbb{C}^p$).

To keep training tractable, neural operators are trained on datasets $\{(\boldsymbol{a}^{(i)}(\boldsymbol{X}_{\mathbb{A}}^{(i)}), \boldsymbol{u}^{(i)}(\boldsymbol{X}_{\mathbb{U}}^{(i)}))\}_{i=1}^n$ consisting of pairs of input and corresponding output functions $(\boldsymbol{a}^{(i)}, \boldsymbol{u}^{(i)}) \in \mathbb{A} \times \mathbb{U}$ that are discretized at finitely many points $\boldsymbol{X}_{\mathbb{A}}^{(i)} \in (\mathbb{D}_{\mathbb{A}})^{n_{\mathbb{A}}^{(i)}}$ and $\boldsymbol{X}_{\mathbb{U}}^{(i)} \in (\mathbb{D}_{\mathbb{U}})^{n_{\mathbb{U}}^{(i)}}$, respectively. The training objective is typically given by the empirical risk

$$R(\boldsymbol{w}) = \frac{1}{n} \sum_{i=1}^n L(\boldsymbol{u}^{(i)}(\boldsymbol{X}_{\mathbb{U}}^{(i)}), \boldsymbol{F}(\boldsymbol{a}^{(i)}(\boldsymbol{X}_{\mathbb{A}}^{(i)}), \boldsymbol{w})(\boldsymbol{X}_{\mathbb{U}}^{(i)}))$$

or a regularized version of the empirical risk. Neural operators were originally developed, and are commonly used, to learn the solution operator of non-linear, parametric partial differential equations. In this case, typically, $\mathbb{D}_{\mathbb{A}} = \mathbb{D}_{\mathbb{U}}$, the input functions $\boldsymbol{a} \in \mathbb{A}$ correspond to parameters and/or initial conditions of the PDE, and the output functions $\boldsymbol{u} \in \mathbb{U}$ are the corresponding solutions of the PDE (at later time points). There are many different realizations of the abstract neural operator framework, including low-rank neural operators (Kovachki et al., 2023), (multipole) graph neural operators (Li et al., 2020b;a), and (spherical) Fourier neural operators (Li et al., 2021; Bonev et al., 2023).

*Example* 2.1 (Fourier Neural Operators). As a case study we will focus on *Fourier neural operators* (FNOs) (Li et al., 2021), a popular variant of the neural operator architecture that applies all spatially global operations in the spectral domain. An FNO $\boldsymbol{F}$ transforms a periodic input function $\boldsymbol{a}$ into a periodic output function $\boldsymbol{F}(\boldsymbol{a}, \boldsymbol{w})(\boldsymbol{x}) :=$

$\boldsymbol{q}(\boldsymbol{v}^{(L)}(\boldsymbol{x}), \boldsymbol{w_q})$ with

$$\boldsymbol{v}_i^{(l+1)}(\boldsymbol{x}) := \sigma^{(l)} \Bigg( \sum_{j=1}^{d'_{\boldsymbol{v}}} \mathcal{F}^{-1} \bigg( \Big( \boldsymbol{R}_{kij}^{(l)} \mathcal{F}\Big(\boldsymbol{v}_j^{(l)}\Big)_k \Big)_{k=1}^{k_{\max}} \bigg)(\boldsymbol{x}) \\ + \boldsymbol{W}_{ij}^{(l)} \boldsymbol{v}_j^{(l)}(\boldsymbol{x}) \Bigg)$$

for $l = 1, \dots, L - 1$ and $\boldsymbol{v}^{(1)}(\boldsymbol{x}) = \boldsymbol{p}(\boldsymbol{a}(\boldsymbol{x}), \boldsymbol{w_p}) \in \mathbb{R}^{d'_{\boldsymbol{v}}}$, where $\mathcal{F}$ denotes the Fourier transform of a periodic function.[1] $\boldsymbol{p} \colon \mathbb{R}^{d'_{\mathbb{A}}} \times \mathbb{W}_{\boldsymbol{p}} \to \mathbb{R}^{d'_{\boldsymbol{v}}}$ and $\boldsymbol{q} \colon \mathbb{R}^{d'_{\boldsymbol{v}}} \times \mathbb{W}_{\boldsymbol{q}} \to \mathbb{R}^{d'_{\mathbb{U}}}$ are parametric functions called *lifting* and *projection*, respectively. $\boldsymbol{R}^{(l)} \in \mathbb{C}^{k_{\max} \times d'_{\boldsymbol{v}} \times d'_{\boldsymbol{v}}}$ and $\boldsymbol{W}^{(l)} \in \mathbb{R}^{d'_{\boldsymbol{v}} \times d'_{\boldsymbol{v}}}$, and $\boldsymbol{w} = (\boldsymbol{w_p}, \boldsymbol{W}^{(1)}, \boldsymbol{R}^{(1)}, \dots, \boldsymbol{W}^{(L-1)}, \boldsymbol{R}^{(L-1)}, \boldsymbol{w_q})$. The map $\boldsymbol{v}^{(l)} \to \boldsymbol{v}^{(l+1)}$ is the $l$-th *Fourier layer*. If the inputs $\boldsymbol{a}$ are discretized on a regular grid, $\mathcal{F}$ can be computed by a real fast Fourier transform (RFFT).

### 2.2. (Gaussian) Random Processes

Aiming to generalize the notion of a Gaussian process later, we provide its formal definition from mathematical statistics that is rarely used in machine learning. For a set $\Omega$ and a set $F$ of functions on $\Omega$ with values in a measurable space, we denote by $\sigma(F)$ the smallest $\sigma$-algebra for which all $f \in F$ are measurable. The *Borel $\sigma$-algebra* on a topological space $\Omega$ is denoted by $\mathcal{B}(\Omega)$. A *random process* on a probability space $(\Omega, \mathcal{A}, \mathrm{P})$ with index set $\mathbb{A}$ and values in a measurable space $(S, \mathcal{A}_S)$ is a function $\mathrm{f} \colon \mathbb{A} \times \Omega \to S$ such that $\mathrm{f}(a, \cdot)$ is $\mathcal{A}$-$\mathcal{A}_S$-measurable for all $a \in \mathbb{A}$. We use the shorthand $\mathrm{f}(a) := \mathrm{f}(a, \cdot)$. One can show that $\omega \mapsto \mathrm{f}(\cdot, \omega)$ is a function-valued random variable with values in $(\mathbb{R}^{\mathbb{A}}, \sigma(\delta_{\mathbb{A}}))$, where $\delta_A$ denotes the set of point evaluation functionals on the set $B^A$ of functions from $A$ to $B$. A random process is called *Gaussian* or a *Gaussian process* (GP) if it has values in $(\mathbb{R}, \mathcal{B}(\mathbb{R}))$ and $\omega \mapsto (\mathrm{f}(a_1, \omega), \dots, \mathrm{f}(a_n, \omega))$ is an $\mathbb{R}^n$-valued Gaussian random variable for all $n \in \mathbb{N}$ and $a_1, \dots, a_n \in \mathbb{A}$. The *mean function* of $\mathrm{f}$ is given by $a \mapsto \mathbb{E}_{\mathrm{P}}[\mathrm{f}(a)]$ and the *covariance function* of $\mathrm{f}$ is given by $(a_1, a_2) \mapsto \mathrm{Cov}_{\mathrm{P}}[\mathrm{f}(a_1), \mathrm{f}(a_2)]$. We denote by $\mathrm{f} \sim \mathcal{GP}(m, k)$ that $\mathrm{f}$ is a Gaussian process with mean function $m$ and covariance function $k$.

It is common to extend the concept of a GP to *finitely* many output dimensions. A $d'$-*output Gaussian process* $\mathbf{f}$ is a random process with values in $(\mathbb{R}^d, \mathcal{B}(\mathbb{R}^d))$ such that $\omega \mapsto (\mathbf{f}(a_1, \omega)^\top \quad \cdots \quad \mathbf{f}(a_n, \omega)^\top)^\top$ is an $\mathbb{R}^{n \cdot d'}$-valued Gaussian random variable for all $n \in \mathbb{N}$, and $a_1, \dots, a_n \in \mathbb{A}$. We use the shorthand $\mathbf{f}(a) := \mathbf{f}(a, \cdot)$. The *mean function* of $\mathbf{f}$ is given by $a \mapsto \mathbb{E}_{\mathrm{P}}[\mathbf{f}(a)] \in \mathbb{R}^{d'}$ and the *covariance function* of $\mathbf{f}$ is $(a_1, a_2) \mapsto \mathrm{Cov}_{\mathrm{P}}[\mathbf{f}(a_1), \mathbf{f}(a_2)] \in \mathbb{R}^{d' \times d'}$. We

---

[1] More precisely, the operator $\mathcal{F} \colon \mathrm{L}_2(\mathbb{T}^d, \mathbb{R}) \to \ell_2(\mathbb{C})$ maps a real-valued square-integrable function on the $d$-dimensional torus to the coefficients of the corresponding Fourier series.

denote by $\mathbf{f} \sim \mathcal{GP}(\boldsymbol{m}, \boldsymbol{K})$ that $\mathbf{f}$ is a multi-output Gaussian process with mean function $\boldsymbol{m}$ and covariance function $\boldsymbol{K}$. While the notion of a multi-output Gaussian process might seem more general than the notion of a Gaussian process, it is possible to "emulate" a function with multiple outputs by augmenting the input space of a Gaussian process:

**Lemma 2.1.** *Let $(\Omega, \mathcal{A}, \mathrm{P})$ be a probability space, $\mathbf{f} \colon \mathbb{A} \times \Omega \to \mathbb{R}^{d'}$, $\mathbb{I} = \{1, \ldots, d'\}$, and $\mathrm{f} \colon (\mathbb{A} \times \mathbb{I}) \times \Omega \to \mathbb{R}$ with $(\mathbf{f}(a, \cdot))_i = \mathrm{f}((a, i), \cdot)$ for all $a \in \mathbb{A}$ and $i \in \mathbb{I}$ (P-almost surely). Then $\mathbf{f} \sim \mathcal{GP}(\boldsymbol{m}, \boldsymbol{K})$ if and only if $\mathrm{f} \sim \mathcal{GP}(m, k)$, where, for all $a \in \mathbb{A}$ and $i \in \mathbb{I}$,*

$$(\boldsymbol{m}(a))_i = m(a, i),$$

*as well as, for all $a_1, a_2 \in \mathbb{A}$ and $i, j \in \mathbb{I}$,*

$$(\boldsymbol{K}(a_1, a_2))_{ij} = k((a_1, i), (a_2, j)).$$

# 3. LUNO: Linearized Predictive Uncertainty in Neural Operators

In this section, we show how to obtain *linearized predictive uncertainty in neural operators* (LUNO). We leverage model linearization to propagate Gaussian weight-space uncertainty through the neural operator to its predictions. LUNO can be applied to trained models as a post-processing step, and does not require expensive re-training. Furthermore, LUNO employs the framework of function-valued Gaussian processes. To that end, we first develop the concept of a function-valued Gaussian process and draw an important parallel with currying in functional programming, which offers a natural interpretation of our method. Figure 1 illustrates the main steps comprising our methodology.

## 3.1. Function-Valued Gaussian Processes and Probabilistic Currying

We want to use model linearization to extend the Gaussian belief over the parameters of a neural network $\boldsymbol{f} \colon \mathbb{R}^d \times \mathbb{R}^p \to \mathbb{R}^{d'}$ into a (multi-output) Gaussian process belief over the function learned by the neural network. However, this is not immediately applicable to neural operators, since their outputs do not lie in $\mathbb{R}^{d'}$, but in a potentially infinite-dimensional Banach space of functions. Hence, we need to generalize (multi-output) Gaussian processes to the notion of a *Banach-valued Gaussian process*.

**Definition 3.1** (Banach-Valued Gaussian Process)**.** Let $\mathbb{U}$ be a real separable Banach space and $\mathbb{L}$ a set[2] of linear functionals on $\mathbb{U}$. A random process $\mathbf{F} \colon \mathbb{A} \times \Omega \to \mathbb{U}$ on a probability space $(\Omega, \mathcal{A}, \mathrm{P})$ with index set $\mathbb{A}$ and values in $(\mathbb{U}, \sigma(\mathbb{L}))$ is called *Gaussian* or a *Gaussian process* if

---

[2]Technically, $\mathbb{L}$ needs to separate the points in $\mathbb{U}$. See Appendix A.1 for additional details.

$\omega \mapsto (\mathbf{F}(a_1, \omega), \ldots, \mathbf{F}(a_n, \omega))$ is a jointly Gaussian random variable[3] for all $n \in \mathbb{N}$ and $a_1, \ldots, a_n \in \mathbb{A}$.

As above, we use the shorthand $\mathbf{F}(a) \coloneqq \mathbf{F}(a, \cdot)$. Moreover, the map $\omega \mapsto \mathbf{F}(\cdot, \omega)$ is a random variable with values in the space of (linear and non-linear) operators $(\mathbb{U}^{\mathbb{A}}, \sigma(\delta_{\mathbb{A}}))$. This warrants the interpretation of $\mathbb{U}$-valued Gaussian processes as *Gaussian random operators*.

In the context of neural operators, $\mathbb{U}$ is a Banach space of $\mathbb{R}^{d'_{\mathbb{U}}}$-valued functions on a common domain $\mathbb{D}_{\mathbb{U}}$. In this case, we can show that $\mathbb{U}$-valued Gaussian processes are closely related to multi-output Gaussian processes with an augmented input space. This is in analogy to Lemma 2.1, but requires some additional technical assumptions.

**Theorem 3.2** (Probabilistic Currying in Banach Spaces; proof in Appendix A.3)**.** *Let $(\Omega, \mathcal{A}, \mathrm{P})$ be a probability space and $\mathbb{U}$ a real separable Banach space of $\mathbb{R}^{d'}$-valued functions with domain $\mathbb{D}_{\mathbb{U}}$. Let $\mathbf{F} \colon \mathbb{A} \times \Omega \to \mathbb{U}$ and $\mathbf{f} \colon (\mathbb{A} \times \mathbb{D}_{\mathbb{U}}) \times \Omega \to \mathbb{R}^{d'}$ such that $\mathbf{F}(a, \cdot)(\boldsymbol{x}) = \mathbf{f}((a, \boldsymbol{x}), \cdot)$ for all $\boldsymbol{a} \in \mathbb{A}$ and $\boldsymbol{x} \in \mathbb{D}_{\mathbb{U}}$ (P-almost surely). Then (i) $\mathbf{F}$ is a random process with values in $(\mathbb{U}, \sigma(\delta_{\mathbb{U}}))$ if and only if $\mathbf{f}$ is a $\mathbb{R}^{d'}$-valued random process, (ii) $\mathbf{F}$ is Gaussian if and only if $\mathbf{f}$ is Gaussian, and (iii) if all evaluation maps $\delta_{\boldsymbol{x}} \colon \mathbb{U} \to \mathbb{R}^{d'}, \boldsymbol{u} \mapsto \boldsymbol{u}(\boldsymbol{x})$ are continuous, then (i) holds for $\mathbf{F}$ with values in $(\mathbb{U}, \mathcal{B}(\mathbb{U}))$.*

Theorem 3.2 reveals an insight into the abstract concept of function-valued Gaussian processes: Function-valued Gaussian processes are equivalent to (multi-output) Gaussian processes with augmented input spaces. This equivalence enables the translation of real-valued GPs, a computationally feasible structure, into infinite-dimensional function-valued objects.

**Probabilistic Currying** We note that Theorem 3.2 constitutes a probabilistic analogue of the concept of *currying* from functional programming (and category theory more generally). The Theorem shows the equivalence of the vector-valued (Gaussian) random function $\mathbf{f} \colon \mathbb{A} \times \mathbb{D}_{\mathbb{U}} \to \mathbb{R}^{d'}$ and the (Gaussian) random operator $\mathbf{F} \colon \mathbb{A} \to (\mathbb{D}_{\mathbb{U}} \to \mathbb{R}^{d'})$ with $\mathbf{F}(\boldsymbol{a})(\boldsymbol{x}) \stackrel{\text{a.s.}}{=} \mathbf{f}(\boldsymbol{a}, \boldsymbol{x})$.

*Example* 3.1 (Currying a Continuous Bivariate Gaussian Process)*.* Let $\mathrm{f} \sim \mathcal{GP}(m, k)$ be a bivariate 2-output Gaussian process with compact index set $\mathbb{X}_1 \times \mathbb{X}_2 \subset \mathbb{R}^2$ on $(\Omega, \mathcal{A}, \mathrm{P})$ with (P-almost surely) continuous paths. For instance, this assumption is fulfilled if $m$ is continuous and $k$ is a multivariate Matérn covariance function (Da Costa et al., 2023). Then $a \mapsto \mathrm{f}(a, \cdot)$ is a function-valued Gaussian process. More precisely, Theorem 3.2 shows that the map $\mathrm{F} \colon \mathbb{X}_1 \times \Omega \to C(\mathbb{X}_2), (a, \omega) \mapsto (x \mapsto \mathrm{f}((a, x), \omega))$ is a $C(\mathbb{X}_2)$-valued Gaussian process with index set $\mathbb{X}_1$.

---

[3]See Definition A.4 and Remark A.5.

Thus, an intuitive way to understand function-valued Gaussian processes is as objects that, when evaluated, return a Gaussian process. Currying can also be used to relate the mean and covariance functions of function-valued or more general vector-valued (Gaussian) random processes, and their counterparts defined on the corresponding multi-output (Gaussian) random process. As this discussion is rather technical, we defer it to Appendix A.3.

Appendix A provides an in-depth explanation of our theoretical framework and contains a plethora of theoretical results on Gaussian processes with values in arbitrary (infinite-dimensional) vector spaces that are not necessarily Banach or function spaces. For example, such results are vital for quantifying uncertainty in neural operators applied to PDEs that only admit weak solutions (see Appendix A.4).

## 3.2. Linearization Turns Neural Operators into Function-Valued Gaussian Processes

We use the notion of function-valued Gaussian processes to develop LUNO. We delineate the key components into different steps, visually represented in Figure 1.

**Step 0**   Let $\boldsymbol{F} \colon \mathbb{A} \times \mathbb{W} \to \mathbb{U} \subset (\mathbb{R}^{d'_{\mathbb{U}}})^{\mathbb{D}_{\mathbb{U}}}$ be a neural operator as in Section 2.1 with $\mathbb{W} = \mathbb{R}^p$.

**Step 1**   By uncurrying $\boldsymbol{F}$, we define the function

$$\boldsymbol{f} \colon (\mathbb{A} \times \mathbb{D}_{\mathbb{U}}) \times \mathbb{W} \to \mathbb{R}^{d'_{\mathbb{U}}}, ((\boldsymbol{a}, \boldsymbol{x}), \boldsymbol{w}) \mapsto \boldsymbol{F}(\boldsymbol{a}, \boldsymbol{w})(\boldsymbol{x}).$$

**Step 2**   We obtain a Gaussian belief $\mathbf{w} \sim \mathcal{N}(\boldsymbol{\mu}, \boldsymbol{\Sigma})$ over the parameters of the network. In Bayesian deep learning, a common way to obtain this Gaussian belief is by placing a Gaussian prior $p(\mathbf{w})$ on the network's parameters and then approximating the posterior distribution given the data $p(\mathbf{w} \mid \mathcal{D})$. Well-established (approximate) inference techniques to obtain the posterior over $\mathbf{w}$ include the Laplace approximation (Ritter et al., 2018; Daxberger et al., 2021a; Immer et al., 2021), variational inference (Graves, 2011; Blundell et al., 2015; Khan et al., 2018), and SWAG (Maddox et al., 2019). Since $\boldsymbol{f}$ has values in $\mathbb{R}^{d'_{\mathbb{U}}}$, following Khan et al. (2019); Immer et al. (2021) and Appendix B, we can linearize the model around $\boldsymbol{\mu}$:

$$\boldsymbol{f}((\boldsymbol{a}, \boldsymbol{x}), \boldsymbol{w}) \approx \boldsymbol{f}_{\boldsymbol{\mu}}^{\mathrm{lin}}((\boldsymbol{a}, \boldsymbol{x}), \boldsymbol{w})$$
$$:= \boldsymbol{f}((\boldsymbol{a}, \boldsymbol{x}), \boldsymbol{\mu}) + \mathrm{D}_{\boldsymbol{w}} \boldsymbol{f}((\boldsymbol{a}, \boldsymbol{x}), \boldsymbol{w})|_{\boldsymbol{\mu}} (\boldsymbol{w} - \boldsymbol{\mu})$$

to arrive at an induced approximate $d'_{\mathbb{U}}$-output Gaussian process belief $\mathbf{f} := \boldsymbol{f}_{\boldsymbol{\mu}}^{\mathrm{lin}}(\cdot, \mathbf{w}) \sim \mathcal{GP}(\boldsymbol{m}, \boldsymbol{K})$ with index set $\mathbb{A} \times \mathbb{D}_{\mathbb{U}}$, $\boldsymbol{m}(\boldsymbol{a}, \boldsymbol{x}) = \boldsymbol{f}((\boldsymbol{a}, \boldsymbol{x}), \boldsymbol{\mu})$, and

$$\boldsymbol{K}((\boldsymbol{a}_1, \boldsymbol{x}_1), (\boldsymbol{a}_2, \boldsymbol{x}_2))$$
$$= \mathrm{D}_{\boldsymbol{w}} \boldsymbol{f}((\boldsymbol{a}_1, \boldsymbol{x}_1), \boldsymbol{w})|_{\boldsymbol{\mu}} \boldsymbol{\Sigma} \, \mathrm{D}_{\boldsymbol{w}} \boldsymbol{f}((\boldsymbol{a}_2, \boldsymbol{x}_2), \boldsymbol{w})|_{\boldsymbol{\mu}}^{\top}.$$

The function $\boldsymbol{f}_{\boldsymbol{\mu}}^{\mathrm{lin}}$ is linear in the weights, but it remains highly nonlinear in the input. Moreover, we have $\boldsymbol{f}_{\boldsymbol{\mu}}^{\mathrm{lin}}(\cdot, \boldsymbol{\mu}) = \boldsymbol{f}(\cdot, \boldsymbol{\mu})$, which means that the mean function of $\mathbf{f}$ matches the prediction of the trained neural operator (before linearization) if we set $\boldsymbol{\mu}$ to the weights $\boldsymbol{w}^{\star}$ found during training.

**Step 3**   Probabilistic currying constructs a Gaussian random operator from $\mathbf{f}$. Namely, we define the function

$$\mathbf{F} \colon \mathbb{A} \times \Omega \to \mathbb{U}, (\boldsymbol{a}, \omega) \mapsto (\boldsymbol{x} \mapsto \mathbf{f}((\boldsymbol{a}, \boldsymbol{x}), \omega)).$$

(For this $\mathbf{F}$ to be well-defined, we need to assume that $\mathbf{f}((\boldsymbol{a}, \cdot), \omega)$ is $\in \mathbb{U}$ for all $\boldsymbol{a} \in \mathbb{A}$. See also Appendix A.4). Theorem 3.2 then shows that $\mathbf{F}$ is a $\mathbb{U}$-valued Gaussian process. Moreover, $\mathbb{E}[\mathbf{F}(\boldsymbol{a})(\boldsymbol{x})] = \boldsymbol{F}(\boldsymbol{a}, \boldsymbol{\mu})(\boldsymbol{x})$, and

$$\mathrm{Cov}[\mathbf{F}(\boldsymbol{a}_1)(\boldsymbol{x}_1), \mathbf{F}(\boldsymbol{a}_2)(\boldsymbol{x}_2)]$$
$$= \mathrm{D}_{\boldsymbol{w}} \boldsymbol{F}(\boldsymbol{a}_1, \boldsymbol{w})(\boldsymbol{x}_1)|_{\boldsymbol{\mu}} \boldsymbol{\Sigma} \, \mathrm{D}_{\boldsymbol{w}} \boldsymbol{F}(\boldsymbol{a}_2, \boldsymbol{w})(\boldsymbol{x}_2)|_{\boldsymbol{\mu}}^{\top}.$$

The entire construction, in particular Theorem 3.2, still applies if $\mathbf{w}$ is not Gaussian. In this case, $\mathbf{f}$ and $\mathbf{F}$ are no longer Gaussian, but the formulae for the mean and covariance functions remain valid. We derive a generalization of the method for general separable Banach spaces $\mathbb{U}$ in Appendix A.4. For instance, this is useful if the functions in $\mathbb{U}$ are not pointwise defined, such as weak solutions of PDEs.

### 3.2.1. CASE STUDY: FOURIER NEURAL GAUSSIAN RANDOM OPERATORS

The exposition so far applies generally to neural operators. For Fourier neural operators, a particularly efficient representation of the function-valued posterior process is available. To simplify the exposition, we focus on the case where the Gaussian belief is restricted to the parameters of the final Fourier block $\boldsymbol{w}_{L-1} := (\boldsymbol{R}^{(L-1)}, \boldsymbol{W}^{(L-1)})$. This is a common approach in the context of last-layer Laplace approximation (Kristiadi et al., 2020). In Appendix C.1, we show that the function-valued GP obtained by applying LUNO in this case takes the form

$$\mathrm{F}(\boldsymbol{a})(\boldsymbol{x}) = \tilde{\boldsymbol{q}}(\boldsymbol{m}_{\mathbf{z}^{(L-1)}}(\boldsymbol{x})) + \Big( \mathrm{D}\tilde{\boldsymbol{q}}(\boldsymbol{m}_{\mathbf{z}^{(L-1)}}(\boldsymbol{x}))$$
$$\cdot (\mathbf{z}^{(L-1)}(\boldsymbol{x}) - \boldsymbol{m}_{\mathbf{z}^{(L-1)}}(\boldsymbol{x})) \Big),$$

i.e., $\mathrm{F}(\boldsymbol{a}) \sim \mathcal{GP}(\boldsymbol{m}_{\boldsymbol{a}}, \boldsymbol{K}_{\boldsymbol{a}})$ with

$$\boldsymbol{m}_{\boldsymbol{a}}(\boldsymbol{x}) = \boldsymbol{F}(\boldsymbol{a}, \boldsymbol{w}^{\star})(\boldsymbol{x}), \quad \text{and}$$
$$\boldsymbol{K}_{\boldsymbol{a}}(\boldsymbol{x}_1, \boldsymbol{x}_2) = \mathrm{D}\tilde{\boldsymbol{q}}(\boldsymbol{m}_{\mathbf{z}^{(L-1)}}(\boldsymbol{x}_1)) \boldsymbol{K}_{\mathbf{z}^{(L-1)}}(\boldsymbol{x}_1, \boldsymbol{x}_2)$$
$$\cdot \mathrm{D}\tilde{\boldsymbol{q}}(\boldsymbol{m}_{\mathbf{z}^{(L-1)}}(\boldsymbol{x}_2))^{\top},$$

where $\mathbf{z}^{(L-1)} \sim \mathcal{GP}(\boldsymbol{m}_{\mathbf{z}^{(L-1)}}, \boldsymbol{K}_{\mathbf{z}^{(L-1)}})$ is a multi-output *parametric* Gaussian process whose moments only depend on $\boldsymbol{v}^{(L-1)}$, $\boldsymbol{\mu}$, and $\boldsymbol{\Sigma}$, and $\tilde{\boldsymbol{q}} = \boldsymbol{q}(\cdot, \boldsymbol{w}_q) \circ \sigma^{(L-1)}$.

There are two practical benefits arising from this representation. First, computing the moments of, and drawing samples from $\mathrm{F}(\boldsymbol{a})$ only needs access to the hidden state $\boldsymbol{v}^{(L-1)}$ of the neural operator. We can thus evaluate the Gaussian process belief at arbitrary output points $\boldsymbol{x} \in \mathbb{D}_{\mathbb{U}}$, without the need to compute more than one (full) forward pass of the neural operator. Secondly, since the Gaussian process belief $\mathrm{F}(\boldsymbol{a})$ over the output function is parametric, we can efficiently sample entire functions from it that can then be lazily evaluated at arbitrary points. This is in contrast to general non-parametric Gaussian processes, where one typically discretizes the GP before drawing samples of the function values at the given finite set of points. Such lazy functional samples can be used e.g. for active experimental design and Bayesian optimization (Wilson et al., 2021).

## 4. Related Work

Azizzadenesheli et al. (2024) provide a comprehensive overview of neural operator architectures. These include graph neural operators (Li et al., 2020a), physics-informed neural operators (Li et al., 2024), multi-wavelet neural operators (Gupta et al., 2021) and the widely used Fourier neural operators (Li et al., 2021). FNOs have gained particular prominence, finding applications across various PDE problems (Pathak et al., 2022; Zhang et al., 2023; Li et al., 2023a; Rashid et al., 2022; Qin et al., 2024; Kossaifi et al., 2023; Bonev et al., 2023). Theoretical foundations for FNOs have been established, with Kovachki et al. (2021) proving their universal approximation capabilities for continuous operators, and Lanthaler et al. (2024) analyzing discretization-induced aliasing errors.

While neural operator architectures have advanced, incorporating uncertainty estimation remains challenging. Recent work has approached this problem from different angles. Garg & Chakraborty (2023) applied variational inference to estimate Bayesian posteriors in DeepONets. More closely related to our work, Magnani et al. (2022) developed uncertainty estimates for graph neural operators using Laplace approximation, though their approach does not extend to FNOs, nor does it consider function space formulations. Kumar et al. (2024) combined Gaussian Process priors with Wavelet Neural Operators, optimizing hyperparameters through negative log-marginal likelihood minimization. Additional Bayesian operator frameworks have been explored by Garg & Chakraborty (2022); Batlle et al. (2024); Zou et al. (2024); Mora et al. (2025).

Function-valued Gaussian processes have been studied in the Hilbert space setting by Owhadi (2023); Batlle et al. (2024). Our approach formulates the theory natively within the context of Banach spaces, as neural operators are defined as mappings between such spaces. In the Appendix we prove that, when restricted to the Hilbert space setting, the

theoretical framework Owhadi (2023); Batlle et al. (2024) embeds into ours.

To generate a probabilistic belief over a neural network's weights, various Bayesian posterior approximation techniques are available. One of the most popular is the Laplace approximation, introduced to deep learning by Mackay (1992), which has gained popularity in the Bayesian deep learning community (Ritter et al., 2018; Daxberger et al., 2021a; Kristiadi et al., 2020; Papamarkou et al., 2024). This is also due to its scalability, achieved through various strategies including using log-posterior Hessian approximations (Ritter et al., 2018; Martens, 2020), treating only a subset of the model probabilistically (Daxberger et al., 2021b), employing linearized Laplace (Foong et al., 2019; Immer et al., 2021), or using scalable Gaussian processes methods (Deng et al., 2022; Ortega et al., 2024). Other Bayesian deep learning methods include variational inference (Graves, 2011; Blundell et al., 2015; Khan et al., 2018; Zhang et al., 2018), Markov Chain Monte Carlo (Neal, 1996; Welling & Teh, 2011; Zhang et al., 2020), SWAG (Maddox et al., 2019), or heuristic methods (Gal & Ghahramani, 2016; Maddox et al., 2019). Finally, a widely used approach for uncertainty quantification in deep learning is ensembles (Lakshminarayanan et al., 2017; Hansen & Salamon, 1990), that train multiple independent neural networks with different random initializations and aggregate the predictions.

## 5. Experiments

We evaluate linearized predictive uncertainty (LUNO-∗) against sample-based approaches (Sample-∗), which require additional approximations to impose a Gaussian Process structure over the output space. To be precise, in the Sample-∗ methods, we draw samples from the weight-space belief, map the samples through the (nonlinear) map $\boldsymbol{w} \mapsto \boldsymbol{F}(\cdot, \boldsymbol{w})$, and compute a function-valued Gaussian process belief over the prediction by moment matching the empirical mean and covariance function. We consider isotropic Gaussian (∗-Iso) and low-rank Laplace approximated (∗-LA) weight-space uncertainties, in both their sample-based (Sample) and linearized (LUNO) forms. We compare these weight-space-Gaussian methods against *input perturbations* (Pathak et al., 2022), and deep ensembles. Deep ensembles were trained 10 times with different random seeds on the original Fourier neural operator (FNO) architecture. We evaluate our model on time-dependent PDEs in one and two spatial dimensions, predicting the next time step autoregressively from the previous ten. We assess uncertainty quantification in two key settings: (1) a low-data regime, where the model is trained on a limited number of trajectories, and (2) out-of-distribution (OOD) scenarios, where physical phenomena unseen during training are introduced at test time.

We evaluate the predictive uncertainty using standard met-

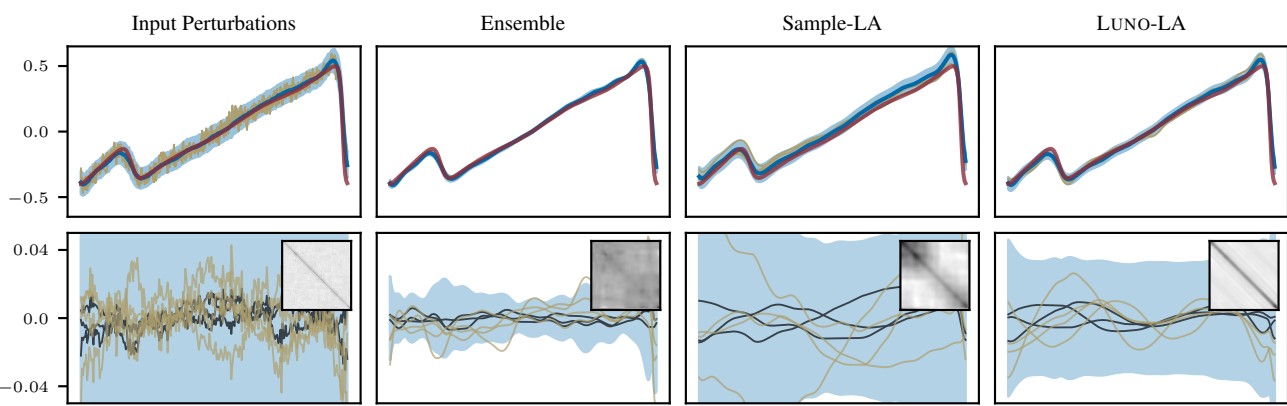

Figure 2: FNO predictive uncertainty quantified by several different methods. Top row: target function (—), mean (—) and 1.96 standard deviations (■) of, as well as samples (—) from, the predictive belief. For the ensemble, the samples are four of the ensemble members. Bottom row: spread of the predictive distribution around the mean. For the sample-/ensemble-based methods, we construct a Gaussian distribution from the empirical covariance matrix and draw four samples (—). We plot 1.96 standard deviations (■) of the predictive belief, as well as the top-three eigenfunctions (—) and a heatmap of the predictive covariance matrix (top right corner of panels).

rics: the expected root mean squared error (RMSE) of the mean predictions, the expected marginal $\chi^2$ statistics, and the expected marginal negative log-likelihood (NLL) over 250 test input-output pairs. Hyperparameters are optimized via grid search using the expected marginal NLL on a validation set as the target. Full details, including data generation, training procedures, uncertainty estimation methods, and more detailed results are provided in the Appendix.

**Code.** We provide an efficient implementation of the LUNO framework in JAX (Bradbury et al., 2018) at

 / MethodsOfMachineLearning / luno.

The code for our experiments can be found at

 / 2bys / luno-experiments.

**Low data regime.** We train an FNO for 100 epochs on 25 simulated solutions of Burgers' equation with 59-time steps and evaluate their uncertainty on 250 unseen test pairs. Figure 2 visualizes the predictive uncertainty for input perturbations, deep ensemble, Sample-LA, and LUNO-LA on a single test data point of Burgers' equation. Table 1 shows that LUNO-LA outperforms the other approaches. This trend holds across two other one-dimensional time-dependent PDE datasets, which are included in the Appendix (Table 4 and Table 5). While all methods produce marginal confidence bands around the network prediction, their sample path covariances differ qualitatively.

**Out-of-Distribution.** To assess OOD robustness, we train an FNO (or an ensemble of 10 FNOs) on a two-dimensional Advection-Diffusion equation with initial conditions sampled from Gaussian blobs and a random constant velocity

| Method | RMSE ($\downarrow$) | $\chi^2$ | NLL ($\downarrow$) |
|---|---|---|---|
| Input Perturbations | $3.63 \times 10^{-2}$ | 0.894 | $-1.8720$ |
| Ensemble | $\mathbf{3.49 \times 10^{-2}}$ | 5.597 | $-0.8145$ |
| Sample-Iso | $3.72 \times 10^{-2}$ | 0.977 | $-1.9341$ |
| LUNO-Iso | $3.62 \times 10^{-2}$ | 0.864 | $-1.9488$ |
| Sample-LA | $5.59 \times 10^{-2}$ | 2.774 | $-1.1572$ |
| LUNO-LA | $3.62 \times 10^{-2}$ | 1.022 | $\mathbf{-2.0787}$ |

Table 1: Comparison of UQ methods for an FNO trained on 25 trajectories of Burgers' equation.

| Method | Base | Flip | Pos-Neg-Flip |
|---|---|---|---|
| Input Perturbations | $-2.586$ | 2.573 | 494.935 |
| Ensemble | $\mathbf{-5.313}$ | $\mathbf{-3.825}$ | $\mathbf{-1.014}$ |
| Sample-Iso | $-2.921$ | 4.071 | 43.362 |
| LUNO-Iso | $-2.892$ | 3.450 | 37.733 |
| Sample-LA | $-2.576$ | 4.395 | 27.046 |
| LUNO-LA | $-2.934$ | $-1.126$ | 1.164 |

Table 2: Expected marginal NLL evaluation across OOD datasets for different methods. Lower is better.

field. We introduce various additional physical phenomena to the test set. These include reversing the velocity field at the center (Flip), introducing a triangular heat source (Pos), and a cloud-shaped heat sink (Neg). Table 2 reports expected marginal NLL over a variation of out-of-distribution datasets. Additional and more granular results can be found in the Appendix. While LUNO-LA outperforms the other weight space methods and input perturbations, deep ensembles achieve the lowest expected marginal NLL in next-step prediction. However, their uncertainty representation is fundamentally different. Figure 3 compares deep ensemble

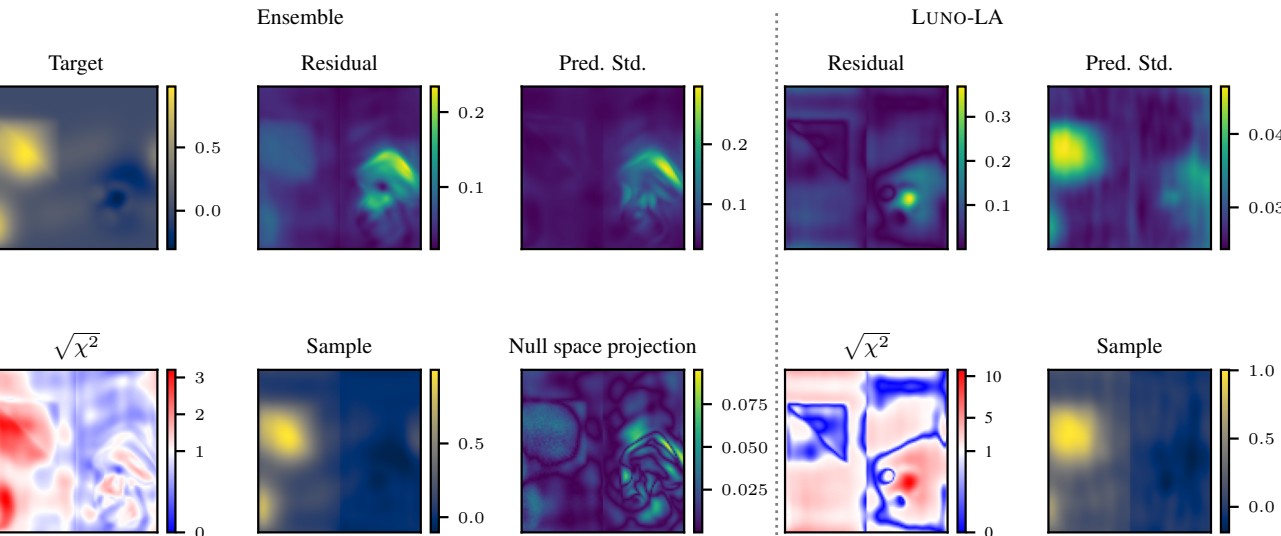

Figure 3: Comparing an ensemble (left), LUNO-LA (right). Top row shows target, residuals, and the predictive standard deviation. Bottom row shows the absolute ratio of the pointwise residual and the predictive standard deviation as well as a sample from the predictive belief. Since the uncertainty structure of the ensemble prediction is of low rank, we also include its unexplained error by projecting the residual vector onto the null space of the predictive covariance.

with LUNO-LA. Deep ensembles approximate uncertainty using a small set of discrete hypotheses, represented by a collection of point masses in parameter space. While this representation is not confined to the analytic form of a Gaussian distribution, it has other constraints: For example, although marginal uncertainty estimates (panel 3 in the figure) can be relatively well-structured, the associated empirical covariance across the ensemble is fundamentally rank-deficient. This limitation is critical, as it leaves certain types of errors entirely unaccounted for (panel 8, which projects residuals onto the null space of the ensemble covariance). By contrast, LUNO-LA constructs a covariance matrix whose rank is (in theory) only bounded by the number of parameters considered.[4] As a result, a plot like panel 8 in Figure 3 does not make sense for LUNO-LA, since, in principle, it explains any variation in the data (albeit with varying calibration). This behavior is also evident in full-trajectory evaluations. Although FNOs are trained for next-step prediction, they are often used for auto-regressive roll-outs, where predictions are recursively fed back as inputs. Such roll-outs cause a subtle yet significant distribution shift, as prediction errors accumulate and are treated as ground truth for subsequent steps. While the deep ensemble improves upon the network prediction in terms of RMSE, its uncertainty estimate does not adapt to the increasing error, as reflected in the NLL (cf. Figure 4).

Wall-clock times for single-trajectory predictions across all

methods are reported in the Appendix. Due to the efficiency of Jacobian-vector products and analytical tractability of the inverse real fast Fourier transform, LUNO-∗ methods outperform their Samples-∗ counterparts, with LUNO-Iso being even faster than the deep ensemble in our implementation. Each method comes with its own additional cost. While deep ensembles need fully separate training runs with different random seeds, LUNO-LA's main computational bottleneck is computing the low-rank approximation of the generalized Gauss–Newton matrix (GGN). This cost is dominated by network size, the selected rank, and the amount of data used for the GGN approximation.

## 6. Conclusion

We introduced LUNO, a framework for predictive uncertainty quantification in neural operators using function-valued Gaussian processes. LUNO can be interpreted as a probabilistic generalization of currying in functional programming. By leveraging model linearization, it offers a computationally efficient and theoretically grounded approach to incorporating weight-space uncertainties in neural operators. The framework endows neural operators with structured weight-space uncertainty quantification capabilities while preserving their resolution-agnostic nature. We demonstrate this for LUNO-LA in the FNO setting under low-data regimes and out-of-distribution scenarios.

LUNO's main limitation lies in the challenges associated with modeling weight-space covariances. Nevertheless, by successfully constructing a structured Gaussian process over

---

[4]Numerical instabilities and (near) singular Jacobians might reduce the rank in practice.

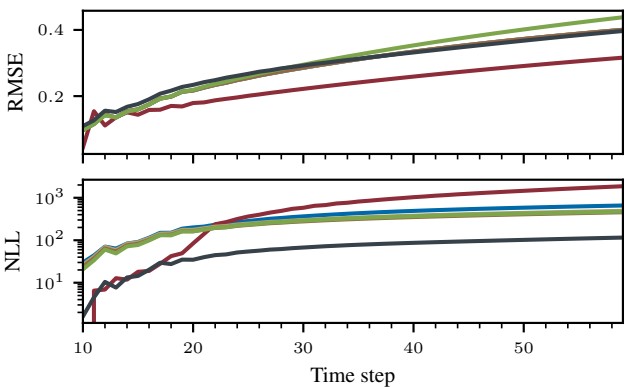

Figure 4: Averaged performance of different UQ methods on an autoregressive rollout of the FNO on 50 trajectories from the Pos-Neg-Flip dataset. We compare input perturbations (—), deep ensembles (—), Sample-Iso (—), LUNO-Iso (—), Sample-LA (—), LUNO-LA (—).

the output space, LUNO paves the way for future applications of GP-valued neural operators in scientific and engineering domains.

## Impact Statement

This paper presents work whose goal is to advance the field of Machine Learning. There are many potential societal consequences of our work, none which we feel must be specifically highlighted here.

## Acknowledgments

The authors gratefully acknowledge financial support by the European Research Council through ERC CoG Action 101123955 ANUBIS ; the DFG Cluster of Excellence "Machine Learning - New Perspectives for Science", EXC 2064/1, project number 390727645; the German Federal Ministry of Education and Research (BMBF) through the Tübingen AI Center (FKZ: 01IS18039A); the DFG SPP 2298 (Project HE 7114/5-1), and the Carl Zeiss Foundation, (project "Certification and Foundations of Safe Machine Learning Systems in Healthcare"), as well as funds from the Ministry of Science, Research and Arts of the State of Baden-Württemberg. The authors thank the International Max Planck Research School for Intelligent Systems (IMPRS-IS) for supporting Emilia Magnani, Marvin Pförtner and Tobias Weber. We would like to thank Nathaël Da Costa for invaluable feedback on the theoretical aspects of this work.

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

# Linearization Turns Neural Operators into Function-Valued Gaussian Processes: Supplementary Materials

## A. Theoretical Results

### A.1. Dual Spaces

We aim to quantify epistemic uncertainty about learned (nonlinear) operators $\boldsymbol{F}\colon \mathbb{A} \to \mathbb{U}$, where $\mathbb{U}$ is a (typically infinite-dimensional) real vector space of functions. If $\mathbb{U}$ is a space of real-valued functions, at the very least, we want to be able to express a probabilistic belief over all point evaluations $\boldsymbol{F}(a)(\boldsymbol{x}) =: \delta_{\boldsymbol{x}}(\boldsymbol{F}(a))$. Note that point evaluation $\delta_{\boldsymbol{x}}\colon \mathbb{U} \to \mathbb{R}$ of functions in $\mathbb{U}$ is a linear map, since $\delta_{\boldsymbol{x}}(\alpha_1 u_1 + \alpha_2 u_2) = (\alpha_1 u_1 + \alpha_2 u_2)(\boldsymbol{x}) = \alpha_1 u_1(\boldsymbol{x}) + \alpha_2 u_2(\boldsymbol{x}) = \alpha_1 \delta_{\boldsymbol{x}}(u_1) + \alpha_2 \delta_{\boldsymbol{x}}(u_2)$. Many interesting operations that map functions into real numbers like (point-evaluated) derivatives and integrals are linear.

Now let $\mathbb{U}$ be an arbitrary real vector space. Real-valued linear maps on $\mathbb{U}$ are referred to as *linear functionals*. The set of all linear functionals on $\mathbb{U}$ is referred to as the *algebraic dual (space)* of $\mathbb{U}$ and denoted by $\mathbb{U}^{\#}$. A subset $\mathbb{L}$ of $\mathbb{U}^{\#}$ is said to be *total* or to *separate the points in* $\mathbb{U}$ if for any $u_1, u_2 \in \mathbb{U}$ with $u_1 \neq u_2$, there is $\ell \in \mathbb{L}$ such that $\ell(u_1) \neq \ell(u_2)$. Such subsets are useful, since they allow us to identify elements from the primal space $\mathbb{U}$ uniquely. For instance, the set of all point evaluation functionals on a vector space of real-valued functions separates the points in the space. If $\mathbb{U}$ is a topological vector space (for instance a separable Banach space in the context of neural operators), then the subspace of continuous linear functionals is denoted by $\mathbb{U}' \subset \mathbb{U}^{\#}$.

*Remark* A.1 (The Bidual Embedding). The algebraic dual space $\mathbb{U}^{\#}$ with pointwise addition and scalar multiplication is a real vector space itself. Hence, any subspace $\mathbb{L} \subset \mathbb{U}^{\#}$, has an algebraic dual space $\mathbb{L}^{\#}$. The elements $\phi \in \mathbb{L}^{\#}$ of this space are linear functions mapping linear functionals into real numbers, i.e., $\phi(\ell) \in \mathbb{R}$ for $\ell \in \mathbb{L} \subset \mathbb{U}^{\#}$. $\mathbb{L}$ is a vector space of real-valued functions, so we can consider its point evaluation functionals $\delta_u\colon \mathbb{L} \to \mathbb{R}, \ell \mapsto \ell(u)$. Note that the map $\iota_{\mathbb{U}, \mathbb{L}^{\#}}\colon \mathbb{U} \to \mathbb{L}^{\#}, u \mapsto \delta_u$ is linear and, if $\mathbb{L}$ separates the points in $\mathbb{U}$, injective. Hence, $\mathbb{U}$ is isomorphic to its image $\delta_{\mathbb{U}} := \iota_{\mathbb{U}, \mathbb{L}^{\#}}(\mathbb{U})$ under $\iota_{\mathbb{U}, \mathbb{L}^{\#}}$. We refer to the map $\iota_{\mathbb{U}, \mathbb{L}^{\#}}$ as the *bidual embedding*. Abusing notation, we write $\mathbb{U} \subset \mathbb{L}^{\#}$, $u \in \mathbb{L}^{\#}$ for $u \in \mathbb{U}$, etc.

### A.2. Probability Measures on Vector Spaces

Our framework models the predictive uncertainty over an output of a neural operator as a random variable with values in an (infinite-dimensional) vector space $\mathbb{U}$ of functions. As noted before, we at least want to quantify the uncertainty about a given set $\mathbb{L} \subset \mathbb{U}^{\#}$ of linear functionals. Hence, we need to make the linear functionals in $\mathbb{L}$ measurable.

Let $(\Omega, \mathcal{A}, \mathrm{P})$ be a probability space, $\mathbb{U}$ a real vector space and $\mathbb{L} \subset \mathbb{U}^{\#}$ a vector subspace of linear functionals separating the points in $\mathbb{U}$. We equip $\mathbb{U}$ with the smallest $\sigma$-algebra $\sigma(\mathbb{L})$ that makes all the functionals in $\mathbb{L}$ measurable. A random variable $\mathrm{u}$ with values in $(\mathbb{U}, \sigma(\mathbb{L}))$ ($\mathbb{U}$-valued for short) is an $\mathcal{A}$-$\sigma(\mathbb{L})$-measurable function $\mathrm{u}\colon \Omega \to \mathbb{U}$.

Similar to their finite-dimensional counterparts, probability measures on and random variables with values in (infinite-dimensional) vector spaces admit the definition of a mean and a (cross-)covariance operator.

**Definition A.2** (Mean and Covariance Operator (see e.g., Bogachev, 1998, Definition 2.2.7)). Let $\gamma$ be a probability measure on $\sigma(\mathbb{L})$.

(a) If $\mathbb{L} \subset \mathrm{L}_1(\gamma)$, then $m_\gamma \in \mathbb{L}^{\#}$ defined by

$$m_\gamma(\ell) := \mathbb{E}_\gamma[\ell] = \int_{\mathbb{U}} \ell(u)\gamma(\mathrm{d}u) \qquad \forall \ell \in \mathbb{L}$$

is called the *mean* of $\gamma$. The mean $m_{\mathrm{u}}$ of a random variable $\mathrm{u}\colon \Omega \to \mathbb{U}$ with values in $(\mathbb{U}, \sigma(\mathbb{L}))$ is defined as the mean $m_{\mathrm{P} \circ \mathrm{u}^{-1}}$ of its law.

(b) If $\mathbb{L} \subset \mathrm{L}_2(\gamma)$, then the linear operator $\mathcal{C}_\gamma \colon \mathbb{L} \to \mathbb{L}^\#$ defined by

$$\mathcal{C}_\gamma(\ell_1)(\ell_2) \coloneqq \mathrm{Cov}_\gamma\left[\ell_1, \ell_2\right] = \int_{\mathbb{U}} \left(\ell_1(u) - m_\gamma(\ell_1)\right)\left(\ell_2(u) - m_\gamma(\ell_2)\right)\gamma(\mathrm{d}u) \qquad \forall \ell_1, \ell_2 \in \mathbb{L}$$

is called the *covariance operator* of $\gamma$. The covariance operator $\mathcal{C}_\mathrm{u}$ of a random variable $\mathrm{u} \colon \Omega \to \mathbb{U}$ with values in $(\mathbb{U}, \sigma(\mathbb{L}))$ is defined as the covariance operator $\mathcal{C}_{\mathrm{P} \circ \mathrm{u}^{-1}}$ of its law.

**Definition A.3** (Cross-Covariance Operator). Let $\mathrm{u}_1, \mathrm{u}_2 \colon \Omega \to \mathbb{U}$ be random variables with values in $(\mathbb{U}, \sigma(\mathbb{L}))$ such that $\mathbb{L} \subset \mathrm{L}_2(\mathbb{U}, \sigma(\mathbb{L}), \mathrm{P} \circ \mathrm{u}_i^{-1})$ for $i = 1, 2$. The operator $\mathcal{C}_{\mathrm{u}_1, \mathrm{u}_2} \colon \mathbb{L} \to \mathbb{L}^\#$ defined by

$$\mathcal{C}_{\mathrm{u}_1, \mathrm{u}_2}(\ell_1)(\ell_2) \coloneqq \mathrm{Cov}\left[\ell_1(\mathrm{u}_1), \ell_2(\mathrm{u}_2)\right] = \int_{\mathbb{U}} \left(\ell_1(\mathrm{u}_1(\omega)) - m_{\mathrm{u}_1}(\ell_1)\right)\left(\ell_2(\mathrm{u}_2(\omega)) - m_{\mathrm{u}_2}(\ell_2)\right)\mathrm{P}(\mathrm{d}\omega)$$

is called the *cross-covariance operator* between $\mathrm{u}_1$ and $\mathrm{u}_2$.

Gaussian measures on $\mathbb{U}$ are defined by generalizing the closure properties of Gaussian measures on $\mathbb{R}^d$.

**Definition A.4** (Gaussian Measure (see e.g., Bogachev, 1998, Definition 2.2.1(a))). A probability measure $\gamma$ on $(\mathbb{U}, \sigma(\mathbb{L}))$ is called *Gaussian* if every linear functional $\ell \in \mathbb{L}$ is a univariate Gaussian random variable on $(\mathbb{U}, \sigma(\mathbb{L}), \gamma)$. A random variable $\mathrm{u} \colon \Omega \to \mathbb{U}$ with values in $(\mathbb{U}, \sigma(\mathbb{L}))$ is called *Gaussian* if its law $\mathrm{P} \circ \mathrm{u}^{-1}$ is Gaussian.

If $\mathbb{L}$ is not a vector space, then we say that a random variable is Gaussian with values in $(\mathbb{U}, \sigma(\mathbb{L}))$ if and only if it is Gaussian with values in $(\mathbb{U}, \sigma(\mathrm{span}\,\mathbb{L}))$. Note that $\sigma(\mathrm{span}\,\mathbb{L}) = \sigma(\mathbb{L})$ for any subset $\mathbb{L} \subset \mathbb{U}^\#$ (by Klenke, 2014, Definition 1.79 and Theorem 1.91).

*Remark* A.5 (Jointly Gaussian Measures). To define Gaussian processes with values in $\mathbb{U}$, we need the notion of a joint Gaussian measure on $\mathbb{U}^n$. Fortunately, we can also leverage Definition A.4 for this. $\mathbb{U}^n$ is a vector space under elementwise addition and scalar multiplication. Its algebraic dual space $(\mathbb{U}^n)^\#$ is isomorphic to $(\mathbb{U}^\#)^n$, i.e. $\ell \in (\mathbb{U}^n)^\#$ if and only if there are $\ell_1, \dots, \ell_n \in \mathbb{U}^\#$ such that

$$\ell(u_1, \dots, u_n) = \sum_{i=1}^n \ell_i(u_i) \qquad \forall u_1, \dots, u_n \in \mathbb{U}.$$

It follows that $\sigma(\mathbb{L}^n) = \sigma(\mathbb{L})^{\otimes n}$, where the latter denotes the product $\sigma$-algebra. Hence, following Definition A.4, we call a probability measure $\gamma$ on $(\mathbb{U}^n, \sigma(\mathbb{L})^{\otimes n})$ *Gaussian* if every $\ell \in \mathbb{L}^n$ is a univariate Gaussian random variable on $(\mathbb{U}^n, \sigma(\mathbb{L})^{\otimes n}, \gamma)$.

*Remark* A.6 (Probability Measures on Separable Banach Spaces). The case where $\mathbb{U}$ is a real separable Banach space is of particular interest in the context of neural operators. In this case, we choose $\mathbb{L} = \mathbb{U}'$, i.e. all linear functionals that are continuous with respect to the norm topology. Then the $\sigma$-algebra $\sigma(\mathbb{U}')$ coincides with the Borel $\sigma$-algebra $\mathcal{B}(\mathbb{U})$ generated by the norm topology (Bogachev, 1998, Theorem A.3.7). For Gaussian random variables $\mathrm{u}$ with values in $(\mathbb{U}, \mathcal{B}(\mathbb{U}))$ (and Gaussian measure on $\mathcal{B}(\mathbb{U})$), the mean $m_\mathrm{u}$ is an element of $\mathbb{U}$ and the covariance operator maps into $\mathbb{U}$ (Bogachev, 1998, Theorem 3.2.3). Moreover, for jointly Gaussian random variables $\mathrm{u}_1, \mathrm{u}_2$ with values in $(\mathbb{U}, \mathcal{B}(\mathbb{U}))$, the cross-covariance operator $\mathcal{C}_{\mathrm{u}_1, \mathrm{u}_2}$ maps from $\mathbb{U}'$ to $\mathbb{U}$ (Bogachev, 1998, Theorem 3.2.4).

### A.3. Random Processes with Values in Vector Spaces

Now we have all the necessary preliminaries to define a Gaussian process with values in $(\mathbb{U}, \sigma(\mathbb{L}))$.

**Definition A.7** (Gaussian Process). A *Gaussian process* with index set $\mathbb{A}$ and values in $(\mathbb{U}, \sigma(\mathbb{L}))$ on $(\Omega, \mathcal{A}, \mathrm{P})$ is a function $\mathrm{F} \colon \mathbb{A} \times \Omega \to \mathbb{U}$ such that $\omega \mapsto (\mathrm{F}(a_1, \omega), \dots, \mathrm{F}(a_n, \omega))$ is a joint, i.e., $(\mathbb{U}^n, \sigma(\mathbb{L})^{\otimes n})$-valued, Gaussian random variable for all $n \in \mathbb{N}$ and $a_1, \dots, a_n \in \mathbb{A}$.

As for real-valued or $\mathbb{R}^d$-valued processes, we can also define mean and covariance functions for random processes with values in arbitrary real vector spaces. However, their definition is more technically involved.

**Definition A.8.** Let $\mathrm{F}$ be a random process with index set $\mathbb{A}$ and values in $(\mathbb{U}, \sigma(\mathbb{L}))$ on $(\Omega, \mathcal{A}, \mathrm{P})$.

(a) If $\mathbb{L} \subset \mathrm{L}_1(\mathrm{P} \circ \mathrm{F}(a, \cdot)^{-1})$ for all $a \in \mathbb{A}$, then the function

$$\mathcal{M} \colon \mathbb{A} \to \mathbb{L}^\#, a \mapsto m_{\mathrm{F}(a, \cdot)}$$

is called the *mean function* of $\mathrm{F}$.

(b) If $\mathbb{L} \subset \mathrm{L}_2(\mathrm{P} \circ \mathrm{F}(a, \cdot)^{-1})$ for all $a \in \mathbb{A}$, then the function

$$\mathcal{K} \colon \mathbb{A} \times \mathbb{A} \to (\mathbb{L} \to \mathbb{L}^{\#}), (a_1, a_2) \mapsto \mathcal{C}_{\mathrm{F}(a_1, \cdot), \mathrm{F}(a_2, \cdot)}$$

is referred to as the *covariance function* of F.

*Remark* A.9 (Moments of Banach-Valued Gaussian Processes). If $\mathbb{U}$ is a separable Banach space, then mean function $\mathcal{M}$ takes values in $\mathbb{U}$ and the covariance function $\mathcal{K}$ takes values in the space of nuclear operators $\mathbb{U}' \to \mathbb{U}$ (Bogachev, 1998, Theorem 3.11.24).

In the following, we aim to establish a correspondence between (Gaussian) random processes with values in $(\mathbb{U}, \sigma(\mathbb{L}))$ and (Gaussian) random processes with values in $(\mathbb{R}, \mathcal{B}(\mathbb{R}))$, which we dub *(generalized) probabilistic currying*. Unlike in Lemma 2.1, we need additional technical assumptions for this to work both ways. Denote by $\mathrm{scl}_{w*}(\hat{\mathbb{L}}) \coloneqq \{\ell \in \mathbb{L} \mid \exists \{\ell_i\}_{i\in\mathbb{N}} \subset \hat{\mathbb{L}} \colon \ell_i \to_{w*} \ell\}$ the weak-* *sequential* closure of a set $\hat{\mathbb{L}} \subset \mathbb{L}$, where $\ell_i \to_{w*} \ell$ if and only if $\ell_i(u) \to \ell(u)$ for all $u \in \mathbb{U}$ (Aliprantis & Border, 2006, Section 5.14).

**Assumption A.10.** Let $\hat{\mathbb{L}} \subset \mathbb{L}$ a set of linear functionals on $\mathbb{U}$ such that there is an $n_{\mathrm{scl}} \in \mathbb{N}_0$ with $\mathrm{scl}_{w*}^{n_{\mathrm{scl}}}(\mathrm{span}\,\hat{\mathbb{L}}) = \mathbb{L}$.

**Theorem A.11** (Generalized Probabilistic Currying). *Let $\hat{\mathbb{L}} \subset \mathbb{L}$ be a set of linear functionals separating the points in $\mathbb{U}$. Let $\mathrm{F} \colon \mathbb{A} \times \Omega \to \mathbb{U}$ and $\mathrm{f} \colon (\mathbb{A} \times \hat{\mathbb{L}}) \times \Omega \to \mathbb{R}$ such that $\ell(\mathrm{F}(a, \omega)) = \mathrm{f}((a, \ell), \omega)$ for all $a \in \mathbb{A}$, $\ell \in \hat{\mathbb{L}}$, and $\mathrm{P}$-almost all $\omega \in \Omega$.*

*(i) If $\mathrm{F}$ is a random process with values in $(\mathbb{U}, \sigma(\mathbb{L}))$, then $\mathrm{f}$ is a random process with values in $(\mathbb{R}, \mathcal{B}(\mathbb{R}))$, and*

*(ii) if $\mathrm{F}$ is Gaussian, then so is $\mathrm{f}$.*

*If Assumption A.10 is satisfied, then the reverse implications hold as well.*

We will need the following generalization of Theorem B.6 from (Pförtner et al., 2022).

**Lemma A.12.** *Let $\hat{\mathbb{L}} \subset \mathbb{L}$ such that Assumption A.10 holds. Then $\hat{\mathbb{L}}$ separates the points in $\mathbb{U}$. Moreover, a function $\mathrm{u} \colon \Omega \to \mathbb{U}$ is*

*(a) $\mathcal{A}$-$\sigma(\mathbb{L})$-measurable if $\ell \circ \mathrm{u}$ is $\mathcal{A}$-$\mathcal{B}(\mathbb{R})$-measurable for all $\ell \in \hat{\mathbb{L}}$,*

*(b) a Gaussian random variable with values in $(\mathbb{U}, \sigma(\mathbb{L}))$ if $(\ell_1 \circ \mathrm{u}, \ldots, \ell_n \circ \mathrm{u})$ is jointly Gaussian for all $n \in \mathbb{N}$ and $\ell_1, \ldots, \ell_n \in \hat{\mathbb{L}}$.*

*Proof.* Define $\{\hat{\mathbb{L}}_n\}_{n=0}^{n_{\mathrm{scl}}}$ with $\hat{\mathbb{L}}_0 \coloneqq \mathrm{span}\,\hat{\mathbb{L}}$ and $\hat{\mathbb{L}}_{n+1} \coloneqq \mathrm{scl}_{w*}(\hat{\mathbb{L}}_n)$. By assumption, $\hat{\mathbb{L}}_{n_{\mathrm{scl}}} = \mathbb{L}$.

Assume that $\hat{\mathbb{L}}$ does not separate the points in $\mathbb{U}$. Then there is $u \in \mathbb{U}$ such that $\ell(u) = 0$ for all $\ell \in \hat{\mathbb{L}}$. We proceed by induction. Pick $\ell \in \hat{\mathbb{L}}_0$. Then there are $\alpha_1, \ldots, \alpha_m \in \mathbb{R}$ and $\ell_1, \ldots, \ell_m \in \hat{\mathbb{L}}$ such that $\ell = \sum_{i=1}^m \alpha_i \ell_i$. Hence,

$$\ell(u) = \sum_{i=1}^m \alpha_i \ell_i(u) = 0.$$

Now assume that $\ell(u) = 0$ for $n < n_{\mathrm{scl}}$ and all $\ell \in \hat{\mathbb{L}}_n$. Fix $\ell \in \hat{\mathbb{L}}_{n+1}$. Then there is $\{\ell_i\}_{i\in\mathbb{N}} \subset \hat{\mathbb{L}}_n$ such that $\ell_i \to_{w*} \ell$. Hence,

$$\ell(u) = \lim_{i\to\infty} \ell_i(u) = 0.$$

All in all, it follows that $\mathbb{L} = \hat{\mathbb{L}}_{n_{\mathrm{scl}}}$ does not separate the points in $\mathbb{U}$, which is a contradiction. Hence $\hat{\mathbb{L}}$ separates the points in $\mathbb{U}$.

(a) We need to show that $\ell \circ \mathrm{u}$ is measurable[5] for all $\ell \in \mathbb{L}$ (Klenke, 2014, Theorem 1.81). We proceed by induction. Let $\ell \in \hat{\mathbb{L}}_0$. Then there are $\alpha_1, \ldots, \alpha_m \in \mathbb{R}$ and $\ell_1, \ldots, \ell_m \in \hat{\mathbb{L}}$ such that $\ell = \sum_{i=1}^m \alpha_i \ell_i$. By assumption, $\ell_i \circ \mathrm{u}$ is measurable for all $i = 1, \ldots, m$. Hence, $\ell$ is measurable by Theorem 1.91 in (Klenke, 2014). Now assume that $\ell \circ \mathrm{u}$ is measurable for all $\ell \in \hat{\mathbb{L}}_n$ with $n < n_{\mathrm{scl}}$. Fix $\ell \in \hat{\mathbb{L}}_{n+1}$. Then there is a sequence $\{\ell_i\}_{i\in\mathbb{N}} \subset \hat{\mathbb{L}}_n$ such that $\ell_i \xrightarrow{w*} \ell$. This implies that $\ell_i \circ \mathrm{u} \to \ell \circ \mathrm{u}$ pointwise, where, by the inductive hypothesis, $\ell_i \circ \mathrm{u}$ is measurable for all $i \in \mathbb{N}$. Hence, $\ell \circ \mathrm{u}$ is measurable for all $\ell \in \hat{\mathbb{L}}_{n+1}$ (Klenke, 2014, Theorem 1.92).

---

[5] We will drop the $\mathcal{A}$-$\mathcal{B}(\mathbb{R})$ prefix for real-valued functions in this proof.

(b) By (a), u is $\mathcal{A}$-$\sigma(\mathbb{L})$-measurable. It suffices to show that u is Gaussian with values in $(\mathbb{U}, \sigma(\hat{\mathbb{L}}_n))$ (Bogachev, 1998, Definition 2.2.1(i)) for all $n = 0, \ldots, n_{\mathrm{scl}}$, which is well-defined, since $\hat{\mathbb{L}}$ separates the points in $\mathbb{U}$. Again, we proceed by induction on $n$. Let $\ell \in \hat{\mathbb{L}}_0$. Then there are $\alpha_1, \ldots, \alpha_m \in \mathbb{R}$ and $\ell_1, \ldots, \ell_m \in \hat{\mathbb{L}}$ such that $\ell = \sum_{i=1}^m \alpha_i \ell_i$. By the closure properties of Gaussians under linear maps, we have that $\ell \circ \mathrm{u}$ is Gaussian. Hence, u is Gaussian with values in $(\mathbb{U}, \sigma(\hat{\mathbb{L}}_0))$. Now assume that u is Gaussian with values in $(\mathbb{U}, \sigma(\hat{\mathbb{L}}_n))$ for $n < n_{\mathrm{scl}}$. Fix $\ell \in \hat{\mathbb{L}}_{n+1}$. Then there is a sequence $\{\ell_i\}_{i\in\mathbb{N}} \subset \hat{\mathbb{L}}_n$ such that $\ell_i \xrightarrow{w*} \ell$. This implies that $\ell_i \circ \mathrm{u} \to \ell \circ \mathrm{u}$ pointwise, where, by the inductive hypothesis, $\ell_i \circ \mathrm{u}$ is Gaussian for all $i \in \mathbb{N}$. Since pointwise limits of Gaussians random variables are Gaussian, $\ell \circ \mathrm{u}$ is Gaussian. Hence, u is Gaussian with values in $(\mathbb{U}, \sigma(\hat{\mathbb{L}}_{n+1}))$.

$\square$

*Proof of Theorem A.11.* $\Rightarrow$ (i) Holds by definition.

(ii) Let $a_1, \ldots, a_n \in \mathbb{A}$ and $\ell_1, \ldots, \ell_n \in \mathbb{L}$. By Remark A.5, the linear functionals

$$\tilde{\ell}_i \colon \mathbb{U}^n \to \mathbb{R}, (u_1, \ldots, u_n) \mapsto \ell_i(u_i)$$

are measurable with respect to $\sigma(\mathbb{L})^{\otimes n}$. Moreover, $\omega \mapsto (\mathrm{F}(a_1, \omega), \ldots, \mathrm{F}(a_n, \omega))$ is Gaussian by assumption. Hence,

$$
\begin{aligned}
\omega \mapsto \left( \tilde{\ell}_i(\mathrm{F}(a_1, \omega), \ldots, \mathrm{F}(a_n, \omega)) \right)_{i=1}^n \\
= (\ell_i(\mathrm{F}(a_i, \omega)))_{i=1}^n \\
= (\mathrm{f}((a_i, \ell_i), \omega))_{i=1}^n
\end{aligned}
$$

is Gaussian with values in $\mathbb{R}^n$.

$\Leftarrow$ (i) Follows directly by applying Lemma A.12(a) to each $\mathrm{F}(a, \cdot)$ individually.

(ii) Let $a_1, \ldots, a_n \in \mathbb{A}$. We have to show that $\omega \mapsto (\mathrm{F}(a_1, \omega), \ldots, \mathrm{F}(a_n, \omega))$ is a Gaussian random variable with values in $\mathbb{U}^n$. It is easy to check that Assumption A.10 holds for $\hat{\mathbb{L}}^n \subset \mathbb{L}^n$. Hence, by Lemma A.12(b), $\omega \mapsto (\mathrm{F}(a_1, \omega), \ldots, \mathrm{F}(a_n, \omega))$ is Gaussian with values in $\mathbb{U}^n$.

$\square$

The following Corollary shows that Assumption A.10 is automatically fulfilled for $\mathbb{L} = \mathbb{U}'$ in separable Banach spaces.

**Corollary A.13** (Generalized Probabilistic Currying in Separable Banach Spaces). *Let $\mathbb{U}$ be a real separable Banach space and $\hat{\mathbb{L}} \subset \mathbb{U}'$ a set of continuous linear functionals separating the points in $\mathbb{U}$. Let $\mathrm{F} \colon \mathbb{A} \times \Omega \to \mathbb{U}$ and $\mathrm{f} \colon (\mathbb{A} \times \hat{\mathbb{L}}) \times \Omega \to \mathbb{R}$ such that $\ell(\mathrm{F}(a, \omega)) = \mathrm{f}((a, \ell), \omega)$ for all $a \in \mathbb{A}$, $\ell \in \hat{\mathbb{L}}$, and $\mathrm{P}$-almost all $\omega \in \Omega$.*

*(i) $\mathrm{F}$ is a random process with values in $(\mathbb{U}, \mathcal{B}(\mathbb{U}))$ if and only if $\mathrm{f}$ is a random process with values in $(\mathbb{R}, \mathcal{B}(\mathbb{R}))$, and*

*(ii) $\mathrm{F}$ is Gaussian if and only if $\mathrm{f}$ is Gaussian.*

*Proof.* We will show that Assumption A.10 is fulfilled for $\mathbb{L} = \mathbb{U}'$ and $n_{\mathrm{scl}} = 1$. Let $\iota \colon \mathbb{U} \to (\hat{\mathbb{L}} \to \mathbb{R}), u \mapsto (\ell \mapsto \ell(u))$, which is linear and injective, since $\hat{\mathbb{L}}$ separates the points in $\mathbb{U}$. Then $\tilde{\mathbb{U}} := \iota(\mathbb{U})$ is isomorphic to $\mathbb{U}$ (as a vector space). Hence, $\tilde{\mathbb{U}}$ equipped with the norm $\|\phi\|_{\tilde{\mathbb{U}}} := \|\iota^{-1}(\phi)\|_{\mathbb{U}}$ is a real separable Banach space which is isometrically isomorphic to $\mathbb{U}$. Moreover, $\tilde{\mathbb{U}}$ is a space of functions with continuous point evaluation functionals, since

$$|\delta_\ell(\iota(u))| = |\ell(u)| \leq \|\ell\|_{\mathbb{U}'} \|u\|_{\mathbb{U}} = \|\ell\|_{\mathbb{U}'} \|\iota(u)\|_{\tilde{\mathbb{U}}}.$$

Thus, there is a sequence $\{\delta_{\ell_i}\}_{i\in\mathbb{N}} \subset \tilde{\mathbb{U}}'$ separating the points in $\mathbb{U}$ (Steinwart, 2024, Theorem 4.10). This implies that $\{\ell_i\}_{i\in\mathbb{N}} \subset \hat{\mathbb{L}}$ separates the points in $\mathbb{U}$. Finally, it follows that $\mathbb{U}' = \mathrm{scl}_{w*}(\mathrm{span}\,\hat{\mathbb{L}})$ (Steinwart, 2024, Proposition 4.3). $\square$

If $\mathbb{U} \subset \mathbb{R}^{\mathbb{D}_{\mathbb{U}}}$ is a vector space of real-valued[6] functions and $\mathbb{L} = \mathrm{span}\,\delta_{\mathbb{D}_{\mathbb{U}}}$, then Theorem A.11 and Corollary A.13 become substantially sharper.

**Corollary A.14** (Probabilistic Currying). *Let $\mathbb{U} \subset \mathbb{R}^{\mathbb{D}_{\mathbb{U}}}$ be a vector space of real-valued functions, $\mathrm{F} \colon \mathbb{A} \times \Omega \to \mathbb{U}$, and $\mathrm{f} \colon (\mathbb{A} \times \mathbb{D}_{\mathbb{U}}) \times \Omega \to \mathbb{R}$ such that $\mathrm{F}(a, \omega)(x) = \mathrm{f}((a, x), \omega)$ for all $a \in \mathbb{A}$, $x \in \mathbb{D}_{\mathbb{U}}$, and $\mathrm{P}$-almost all $\omega \in \Omega$. Then*

*(i) $\mathrm{F}$ is a random process with values in $(\mathbb{U}, \sigma(\delta_{\mathbb{D}_{\mathbb{U}}}))$ if and only if $\mathrm{f}$ is a random process with values in $(\mathbb{R}, \mathcal{B}(\mathbb{R}))$,*

*(ii) $\mathrm{F}$ has a mean function $\mathcal{M}$ with values in $\mathbb{U}$ if and only if $\mathrm{f}$ has a mean function $m$, where $\mathcal{M}(a) = m(a, \,\cdot\,)$ for all $a \in \mathbb{A}$,*

*(iii) $\mathrm{F}$ has a covariance function $\mathcal{K}$ with values in $\mathrm{span}\,\delta_{\mathbb{D}_{\mathbb{U}}} \to \mathbb{U}$ if and only if $\mathrm{f}$ has a covariance function $k$, where $\mathcal{K}(a_1, a_2)(\delta_x) = k((a_1, x), (a_2, \,\cdot\,))$ for all $a_1, a_2 \in \mathbb{A}$, and $x \in \mathbb{D}_{\mathbb{U}}$, and*

*(iv) $\mathrm{F}$ is Gaussian if and only if $\mathrm{f}$ is Gaussian.*

*If $\mathbb{U}$ is a separable Banach space with continuous point evaluation functionals, then*

*(v) (i) and (iv) hold for $\mathrm{F}$ with values in $(\mathbb{U}, \mathcal{B}(\mathbb{U}))$, and,*

*(vi) if it exists, then the covariance function $\mathcal{K}$ in (iii) has values in $\mathbb{U}' \to \mathbb{U}$, where*

$$\mathcal{K}(a_1, a_2)(\ell)(x) = \ell(k((a_1, \,\cdot\,), (a_2, x)))$$

*for all $\ell \in \mathbb{U}'$ and $x \in \mathbb{D}_{\mathbb{U}}$.*

*Proof.* Follows from Theorem A.11 and Corollary A.13. □

Finally, Theorem 3.2 from the main text is merely a corollary of the results developed above.

**Theorem 3.2** (Probabilistic Currying in Banach Spaces; proof in Appendix A.3). *Let $(\Omega, \mathcal{A}, \mathrm{P})$ be a probability space and $\mathbb{U}$ a real separable Banach space of $\mathbb{R}^{d'}$-valued functions with domain $\mathbb{D}_{\mathbb{U}}$. Let $\mathbf{F} \colon \mathbb{A} \times \Omega \to \mathbb{U}$ and $\mathbf{f} \colon (\mathbb{A} \times \mathbb{D}_{\mathbb{U}}) \times \Omega \to \mathbb{R}^{d'}$ such that $\mathbf{F}(\boldsymbol{a}, \,\cdot\,)(\boldsymbol{x}) = \mathbf{f}((\boldsymbol{a}, \boldsymbol{x}), \,\cdot\,)$ for all $\boldsymbol{a} \in \mathbb{A}$ and $\boldsymbol{x} \in \mathbb{D}_{\mathbb{U}}$ ($\mathrm{P}$-almost surely). Then (i) $\mathbf{F}$ is a random process with values in $(\mathbb{U}, \sigma(\delta_{\mathbb{U}}))$ if and only if $\mathbf{f}$ is a $\mathbb{R}^{d'}$-valued random process, (ii) $\mathbf{F}$ is Gaussian if and only if $\mathbf{f}$ is Gaussian, and (iii) if all evaluation maps $\delta_{\boldsymbol{x}} \colon \mathbb{U} \to \mathbb{R}^{d'}, \boldsymbol{u} \mapsto \boldsymbol{u}(\boldsymbol{x})$ are continuous, then (i) holds for $\mathbf{F}$ with values in $(\mathbb{U}, \mathcal{B}(\mathbb{U}))$.*

*Proof.* Follows from Corollary A.14 and Lemma 2.1. □

## A.4. Banach-Valued Gaussian Processes from Linearized Neural Operators

For simplicity of the exposition, we limited the construction of the LUNO framework in Section 3.2 to neural operators, which map into a Banach space of functions with continuous point evaluation functionals. This limits its applicability, especially for solving PDEs, whose solutions are often not defined pointwise, but rather elements of Sobolev spaces $W^{p,k}(\mathbb{D}_{\mathbb{U}}) \subset \mathrm{L}^p(\mathbb{D}_{\mathbb{U}})$. Hence, in this section, we extend LUNO to neural operators that map into abstract separable Banach[7] spaces $\mathbb{U}$.

**Step 0** Let $\mathbb{U}$ be a real separable Banach space, $\mathbb{A}$ a set, $\mathbb{W} \subset \mathbb{R}^p$ a subspace, and $F \colon \mathbb{A} \times \mathbb{W} \to \mathbb{U}$ a neural operator.

**Step 1** First, we select a subset $\hat{\mathbb{L}} \subset \mathbb{U}'$ of continuous linear functionals separating the points in $\mathbb{U}$, for which we want to quantify predictive uncertainty under the neural operator. Define

$$f \colon (\mathbb{A} \times \hat{\mathbb{L}}) \times \mathbb{W} \to \mathbb{R}, ((a, \ell), \boldsymbol{w}) \mapsto \ell(F(a, \boldsymbol{w})).$$

This is an uncurried version of the neural operator. To see this, note that a neural operator $F \colon \mathbb{A} \times \mathbb{W} \to \mathbb{U}$ can be uniquely identified with the function

$$\tilde{F} \colon \mathbb{A} \times \mathbb{W} \to (\hat{\mathbb{L}} \to \mathbb{R}), (a, \boldsymbol{w}) \mapsto (\ell \mapsto \ell(F(a, \boldsymbol{w}))).$$

---

[6]By a generalization of Lemma 2.1, an analogous result holds in vector spaces of $\mathbb{R}^{d'}$-valued functions.

[7]With minor modifications, the construction below works in arbitrary real vector spaces.

*Proof.* Let $F_1, F_2\colon \mathbb{A} \times \mathbb{W} \to \mathbb{U}$ be neural operators with $F_1 \neq F_2$. Then there are $a \in \mathbb{A}$ and $\boldsymbol{w} \in \mathbb{W}$ such that $F_1(a, \boldsymbol{w}) \neq F_2(a, \boldsymbol{w})$. Since $\hat{\mathbb{L}}$ separates the points in $\mathbb{U}$, this implies that there is $\ell \in \hat{\mathbb{L}}$ such that $\tilde{F}_1(a, \boldsymbol{w})(\ell) :=$ $F_1(a, \boldsymbol{w}) \neq F_2(a, \boldsymbol{w}) =: \tilde{F}_2(a, \boldsymbol{w})(\ell)$. Hence, $\tilde{F}_1 \neq \tilde{F}_2$. $\qquad\square$

**Step 2**   We model the uncertainty over the parameters as a random variable $\mathbf{w}\colon \Omega \to \mathbb{W}$ on a probability space $(\Omega, \mathcal{A}, \mathrm{P})$ with $\mathrm{supp}(\mathbf{w}) = \mathbb{W}$. As in Section 3.2, we will now linearize $f$ in $\boldsymbol{w}$ around a point $\boldsymbol{w}_0 \in \mathbb{W}$. To achieve this, we assume that the directional derivatives

$$\partial_{\boldsymbol{w}} f((a, \ell), \,\cdot\,)\,(\boldsymbol{w}_0) = \lim_{h \to 0} \frac{f((a, \ell), \boldsymbol{w}_0 + h\boldsymbol{w}) - f((a, \ell), \boldsymbol{w}_0)}{h}$$

at $\boldsymbol{w}_0$ exist for all $a \in \mathbb{A}$, $\ell \in \hat{\mathbb{L}}$, and $\boldsymbol{w} \in \mathbb{W}$, and are linear in $\boldsymbol{w}$. For instance, this is the case if $f((a, \ell), \,\cdot\,)$ is differentiable at $\boldsymbol{w}_0$. In this case, the linearization of $f$ is given by

$$f((a, \ell), \boldsymbol{w}) \approx f_{\boldsymbol{\mu}}^{\mathrm{lin}}((a, \ell), \boldsymbol{w}) := f((a, \ell), \boldsymbol{w}_0) + \partial_{\boldsymbol{w} - \boldsymbol{w}_0} f((a, \ell), \,\cdot\,)\,(\boldsymbol{w}_0)\,.$$

Then the function

$$\mathrm{f}\colon (\mathbb{A} \times \hat{\mathbb{L}}) \times \Omega \to \mathbb{R}, ((a, \ell), \omega) \mapsto f_{\boldsymbol{\mu}}^{\mathrm{lin}}((a, \ell), \mathbf{w}(\omega))$$

is a random process. If $\mathbf{w}$ has a mean $\boldsymbol{\mu}$, then $\mathrm{f}$ has a mean function

$$m\colon \mathbb{A} \times \hat{\mathbb{L}} \to \mathbb{R}, (a, \ell) \mapsto f((a, \ell), \boldsymbol{w}_0) + \partial_{\boldsymbol{\mu} - \boldsymbol{w}_0} f((a, \ell), \,\cdot\,)\,(\boldsymbol{w}_0)\,,$$

and if $\mathbf{w}$ has a covariance matrix $\boldsymbol{\Sigma}$, then $\mathrm{f}$ has a covariance function given by

$$k\colon (\mathbb{A} \times \hat{\mathbb{L}}) \times (\mathbb{A} \times \hat{\mathbb{L}}) \to \mathbb{R}, ((a_1, \ell_1), (a_2, \ell_2)) \mapsto \sum_{i,j=1}^{p} \partial_i f((a, \ell_1), \,\cdot\,)(\boldsymbol{w}_0) \boldsymbol{\Sigma}_{ij} \partial_j f((a, \ell_2), \,\cdot\,)(\boldsymbol{w}_0).$$

Note that $\mathrm{f}$ and $m$ are linear in $\ell$ and $k$ is bilinear in $(\ell_1, \ell_2)$. Moreover, if $\mathbf{w}$ is Gaussian, then $\mathrm{f}$ is a Gaussian process.

**Step 3**   Finally, we construct a $\mathbb{U}$-valued (Gaussian) random process $\mathrm{F}\colon \mathbb{A} \times \Omega \to \mathbb{U}$ by probabilistically currying $\mathrm{f}$. However, this is more challenging for an abstract $\mathbb{U}$. Intuitively, we want to undo the uncurrying operation from Step 1. To this end, we assume[8] that there is $\mathrm{F}\colon \mathbb{A} \times \Omega \to \mathbb{U}$ with $\ell(\mathrm{F}(a, \omega)) = \mathrm{f}((a, \ell), \omega)$ for all $a \in \mathbb{A}$, $\ell \in \hat{\mathbb{L}}$, and P-almost all $\omega \in \Omega$. In this case, Corollary A.13 ensures that

(i)  $\mathrm{F}$ is a random process with values in $(\mathbb{U}, \mathcal{B}\,(\mathbb{U}))$,

(ii)  $\mathrm{F}$ has a mean function $\mathcal{M}\colon \mathbb{A} \to \mathbb{U}$ with

$$\ell(\mathcal{M}(a)) = m(a, \ell) = f((a, \ell), \boldsymbol{w}_0) + \partial_{\boldsymbol{\mu} - \boldsymbol{w}_0} f((a, \ell), \,\cdot\,)\,(\boldsymbol{w}_0)$$

if $\mathbf{w}$ has a mean vector $\boldsymbol{\mu}$,

(iii)  $\mathrm{F}$ has a covariance function $\mathcal{K}\colon \mathbb{A} \times \mathbb{A} \to (\mathbb{U}' \to \mathbb{U})$ with

$$\ell_2(\mathcal{K}(a_1, a_2)(\ell_1)) = k((a_1, \ell_1), (a_2, \ell_2)) = \sum_{i,j=1}^{p} \partial_i f((a, \ell_1), \,\cdot\,)(\boldsymbol{w}_0) \boldsymbol{\Sigma}_{ij} \partial_j f((a, \ell_2), \,\cdot\,)(\boldsymbol{w}_0)$$

if $\mathbf{w}$ has a covariance matrix $\boldsymbol{\Sigma}$, and

(iv)  $\mathrm{F}$ is a Gaussian process if $\mathbf{w}$ is Gaussian.

---

[8] See Appendices A.4.1 and A.4.2 for more details on this assumption.

### A.4.1. WEAK GÂTEAUX DIFFERENTIABILITY

The existence of $F\colon \mathbb{A} \times \Omega \to \mathbb{U}$ with $\ell(F(a,\omega)) = f((a,\ell),\omega)$ for all $a \in \mathbb{A}$, $\ell \in \hat{\mathbb{L}}$, and P-almost all $\omega \in \Omega$ is equivalent to $F(a,\,\cdot\,)$ being $\tau(\mathbb{U}, \hat{\mathbb{L}}^{\#})$-*Gâteaux differentiable at* $\boldsymbol{w}_0$, i.e., there is a linear operator $\delta F(a,\,\cdot\,)(\boldsymbol{w}_0)\colon \mathbb{W} \to \mathbb{U}$ such that

$$\delta F(a,\,\cdot\,)(\boldsymbol{w}_0)(\boldsymbol{w}) = \lim_{h \to 0} \frac{F(a, \boldsymbol{w}_0 + h\boldsymbol{w}) - F(a, \boldsymbol{w}_0)}{h} \qquad \text{in } (\mathbb{U}, \tau(\mathbb{U}, \hat{\mathbb{L}}^{\#})) \tag{A.1}$$

for all $\boldsymbol{w} \in \mathbb{W}$, where $\tau(\mathbb{U}, \hat{\mathbb{L}}^{\#})$ is the smallest topology on $\mathbb{U}$ for which all functionals in $\hat{\mathbb{L}}$ are continuous.

*Proof.* Note that, since $\mathbb{W}$ is a metric space, we can take sequential limits in Equation (A.1) without loss of generality.

$\Rightarrow$ Fix $a \in \mathbb{A}$. We will constuct the Gâteaux derivative from $\tilde{F}(a,\omega)$. Since $\tilde{F}(a,\omega) \in \iota_{\mathbb{U},\hat{\mathbb{L}}^{\#}}(\mathbb{U})$ for P-almost all $\omega \in \Omega$, there is $N \in \mathcal{A}$ with $P(N) = 0$ and $\tilde{F}(a,\omega) \in \iota_{\mathbb{U},\hat{\mathbb{L}}^{\#}}(\mathbb{U})$ for all $\omega \in \Omega \setminus N$. We have $\text{supp}\,\mathbf{w} = \mathbb{W}$ and hence there are $\omega_1, \ldots, \omega_d \in \Omega$ such that $\{\boldsymbol{b}_i\}_{i=1}^d$ with $\boldsymbol{b}_i := \mathbf{w}(\omega_i) - \boldsymbol{w}_0$ is a basis of $\mathbb{W}$. Let $\lambda\colon \mathbb{W} \to \mathbb{U}$ be the unique linear operator with

$$\begin{aligned}
\lambda(\boldsymbol{b}_i) &:= \iota_{\mathbb{U},\hat{\mathbb{L}}^{\#}}^{-1}(\tilde{F}(a,\omega_i)) - F(a, \boldsymbol{w}_0)\\
&= \iota_{\mathbb{U},\hat{\mathbb{L}}^{\#}}^{-1}(f((a,\,\cdot\,),\omega_i) - f((a,\,\cdot\,),\boldsymbol{w}_0))\\
&= \iota_{\mathbb{U},\hat{\mathbb{L}}^{\#}}^{-1}(\ell \mapsto \partial_{\mathbf{w}(\omega_i)-\boldsymbol{w}_0} f((a,\ell),\,\cdot\,)(\boldsymbol{w}_0))\\
&= \iota_{\mathbb{U},\hat{\mathbb{L}}^{\#}}^{-1}(\ell \mapsto \partial_{\boldsymbol{b}_i} f((a,\ell),\,\cdot\,)(\boldsymbol{w}_0)),
\end{aligned}$$

i.e. $\lambda(\boldsymbol{w}) = \iota_{\mathbb{U},\hat{\mathbb{L}}^{\#}}^{-1}(\ell \mapsto \partial_{\boldsymbol{w}} f((a,\ell),\,\cdot\,)(\boldsymbol{w}_0))$ forall $\boldsymbol{w} \in \mathbb{W}$. Since $\mathbb{W}$ is finite dimensional, $\lambda$ is $\tau_{\mathbb{W}}$-$\tau(\mathbb{U}, \hat{\mathbb{L}}^{\#})$-continuous, where $\tau_{\mathbb{W}}$ is the norm topology on $\mathbb{W}$. Let $\{h_n\}_{n \in \mathbb{N}} \subset \mathbb{R}$ be any null sequence. Then for any $\ell \in \hat{\mathbb{L}}$ we have

$$\begin{aligned}
\ell\left( \frac{F(a, \boldsymbol{w}_0 + h_n\boldsymbol{w}) - F(a, \boldsymbol{w}_0)}{h_n} \right) &= \frac{f((a,\ell), \boldsymbol{w}_0 + h_n\boldsymbol{w}) - f((a,\ell), \boldsymbol{w}_0)}{h_n}\\
&\xrightarrow{n \to \infty} \partial_{\boldsymbol{w}} f((a,\ell),\,\cdot\,)(\boldsymbol{w}_0) &= \ell(\lambda(\boldsymbol{w})).
\end{aligned}$$

Hence, $F(a,\,\cdot\,)$ is $\tau(\mathbb{U}, \hat{\mathbb{L}}^{\#})$-Gâteaux differentiable at $\boldsymbol{w}_0$ with Gâteaux derivative $\delta F(a,\,\cdot\,)(\boldsymbol{w}_0) = \lambda$.

$\Leftarrow$ If $F(a,\,\cdot\,)$ is $\tau(\mathbb{U}, \hat{\mathbb{L}}^{\#})$-Gâteaux differentiable at $\boldsymbol{w}_0$ for all $a \in \mathbb{A}$, then we can construct F as

$$F\colon \mathbb{A} \times \Omega \to \mathbb{U}, (a,\omega) \mapsto F(a, \boldsymbol{w}_0) + \delta F(a,\,\cdot\,)(\boldsymbol{w}_0)(\mathbf{w}(\omega) - \boldsymbol{w}_0) \tag{A.2}$$

Then

$$\begin{aligned}
\ell(F(a,\omega)) &= \ell(F(a, \boldsymbol{w}_0)) + \partial_{\mathbf{w}(\omega)-\boldsymbol{w}_0} \ell(F(a,\,\cdot\,))(\boldsymbol{w}_0)\\
&= f((a,\ell), \boldsymbol{w}_0) + \partial_{\mathbf{w}(\omega)-\boldsymbol{w}_0} f((a,\ell),\,\cdot\,)(\boldsymbol{w}_0)\\
&= f_{\boldsymbol{\mu}}^{\text{lin}}((a,\ell), \mathbf{w}(\omega))\\
&= f((a,\ell),\omega)
\end{aligned}$$

for all $a \in \mathbb{A}$, $\ell \in \hat{\mathbb{L}}$, and $\omega \in \Omega$.

$\square$

A stronger condition, namely (norm) Fréchet differentiability, has been verified for Fourier neural operators mapping between $L_p$ spaces (Kabri et al., 2023).

Note that we can also use the properties of the Gâteaux derivative to verify the conclusions of Corollary A.13. Since $\mathbb{W}$ is finite-dimensional, the Gâteaux derivative $\delta F(a,\,\cdot\,)(\boldsymbol{w}_0)$ is continuous with respect to any TVS topology on $\mathbb{U}$. Hence, F

as defined in Equation (A.2) is a random process with values in $(\mathbb{U}, \sigma(\mathbb{U}')) = (\mathbb{U}, \mathcal{B}(\mathbb{U}))$. If $\mathbf{w}$ has a mean $\boldsymbol{\mu}$, then the mean function $\mathcal{M} \colon \mathbb{A} \to \mathbb{U}$ of F is given by

$$\mathcal{M}(a) = F(a, \boldsymbol{w}_0) + \delta F(a, \,\cdot\,)(\boldsymbol{w}_0)(\boldsymbol{\mu} - \boldsymbol{w}_0),$$

and, if $\mathbf{w}$ has a covariance matrix $\boldsymbol{\Sigma}$, then the covariance function $\mathcal{K} \colon \mathbb{A} \times \mathbb{A} \to (\mathbb{U}' \to \mathbb{U})$ of F is given by

$$\mathcal{K}(a_1, a_2) = \delta F(a_1, \,\cdot\,)(\boldsymbol{w}_0)\,\boldsymbol{\Sigma}\,\delta F(a_2, \,\cdot\,)(\boldsymbol{w}_0)'.$$

By the closure properties of Banach-valued Gaussian random variables under continuous affine maps (Bogachev, 1998, Lemma 2.2.2), F is a Gaussian process if $\mathbf{w}$ is Gaussian.

While this construction is somewhat more direct, the currying approach outlined above mimics more closely how F is constructed on a computer, especially when $\mathbb{U} \subset (\mathbb{R}^{d'_\mathbb{U}})^{\mathbb{D}_\mathbb{U}}$ and $\hat{\mathbb{L}} = \{\delta_{\boldsymbol{x}, i} \colon \boldsymbol{x} \in \mathbb{D}_\mathbb{U}, i = 1, \ldots, d'_\mathbb{U}\}$ as in Section 3.2, and does not require knowledge of Banach-valued derivatives.

### A.4.2. BIDUAL RANDOM PROCESSES

If there is no $F \colon \mathbb{A} \times \Omega \to \mathbb{U}$ with $\ell(F(a, \omega)) = f((a, \ell), \omega)$ for all $a \in \mathbb{A}$, $\ell \in \hat{\mathbb{L}}$, and P-almost all $\omega \in \Omega$, we can construct a weaker version of F. Note that, if $\ell = \alpha_1 \ell_1 + \alpha_2 \ell_2 \in \hat{\mathbb{L}}$, then $f((a, \ell), \omega) = \alpha_1 f((a, \ell_1), \omega) + \alpha_2 f((a, \ell_2), \omega)$. This means that $f((a, \,\cdot\,), \omega)$ can always be uniquely linearly extended to $\mathbb{L} = \operatorname{span} \hat{\mathbb{L}}$. Define

$$\tilde{F} \colon \mathbb{A} \times \Omega \to \mathbb{L}^{\#}, (a, \omega) \mapsto (\ell \mapsto f((a, \ell), \omega)).$$

We can use Corollary A.14 to show that $\tilde{F}$ is a random process with values in the algebraic dual $\mathbb{L}^{\#}$ of $\mathbb{L}$ equipped with the smallest $\sigma$-algebra $\sigma(\delta_{\mathbb{L}^{\#}})$ that makes all point evaluation functionals measurable. We refer to such random processes as *bidual random processes*.

Recall the bidual embedding $\iota_{\mathbb{U}, \mathbb{L}^{\#}}$ from Remark A.1. If F exists, then $\tilde{F}(a, \omega) = \iota_{\mathbb{U}, \mathbb{L}^{\#}}(F(a, \omega))$.

### A.5. Operator-Valued Gaussian Processes as Hilbert-Valued Gaussian Processes

Finally, we show that *operator-valued Gaussian processes* (Owhadi, 2023; Batlle et al., 2024; Mora et al., 2025) can be embedded in the theoretical framework outlined above. Since operator-valued Gaussian processes are only defined on separable Hilbert spaces, in this section, we let $\mathbb{A}, \mathbb{U}$ be separable Hilbert spaces with inner products $\langle \cdot, \cdot \rangle_\mathbb{A}$ and $\langle \cdot, \cdot \rangle_\mathbb{U}$, respectively. In this case, the continuous dual $\mathbb{U}'$ is identified with the primal space $\mathbb{U}$ via the Riesz isomorphism. We start by reviewing the building blocks of operator-valued Gaussian processes.

**Definition A.15** (Operator-Valued Kernel (Owhadi, 2023, Definition 9.1)). A function $\mathcal{K} \colon \mathbb{A} \times \mathbb{A} \to (\mathbb{U} \to \mathbb{U})$ is called an *operator-valued kernel* if $\mathcal{K}(a_1, a_2)$ is a bounded linear operator with $\mathcal{K}(a_1, a_2) = \mathcal{K}(a_2, a_1)^*$ for all $a_1, a_2 \in \mathbb{A}$ and $\sum_{i,j=1}^m \langle u_i, \mathcal{K}(a_i, a_j) u_j \rangle_\mathbb{U} \geq 0$ for all $m \in \mathbb{N}$, $a_1, \ldots, a_m \in \mathbb{A}$, and $u_1, \ldots, u_m \in \mathbb{U}$.

**Definition A.16** (Gaussian Hilbert Space (Owhadi & Scovel, 2019, Definition 7.1)). A closed subspace $\mathbb{H}$ of $L_2(\Omega, \mathcal{A}, P)$ is called a *Gaussian Hilbert space* if every $h \in \mathbb{H}$ is a univariate Gaussian random variable.

**Definition A.17** (Operator-Valued Gaussian Process (Owhadi, 2023, Definition 5.1)). Let $\mathcal{M} \colon \mathbb{A} \to \mathbb{U}$, $\mathcal{K} \colon \mathbb{A} \times \mathbb{A} \to (\mathbb{U} \to \mathbb{U})$ an operator-valued kernel, and $\mathbb{H}$ a Gaussian Hilbert space over $(\Omega, \mathcal{A}, P)$. A function $\Xi \colon \mathbb{A} \to (\mathbb{U} \to \mathbb{H})$ is called an *operator-valued Gaussian process* if $\Xi(a)$ is a bounded linear operator for all $a \in \mathbb{A}$. $\Xi$ is said to have mean $\mathcal{M}$ and covariance kernel $\mathcal{K}$ if $\Xi(a)(u) \sim \mathcal{N}(\langle u, \mathcal{M}(a) \rangle_\mathbb{U}, \langle u, \mathcal{K}(a, a)(u) \rangle_\mathbb{U})$ for all $a \in \mathbb{A}$ and $u \in \mathbb{U}$, and

$$\operatorname{Cov}_P [\Xi(a_1)(u_1), \Xi(a_2)(u_2)] = \langle u_1, \mathcal{K}(a_1, a_2)(u_2) \rangle_\mathbb{U}.$$

Generally, operator-valued Gaussian processes are "equivalent to" a subset of Gaussian processes with values in $(\mathbb{U}')^{\#} = \mathbb{U}^{\#}$ (see Appendix A.4.2).

**Proposition A.18.** *Let $\tilde{F} \colon \mathbb{A} \times \Omega \to \mathbb{U}^{\#}$, $\mathbb{H} \subset L_2(\Omega, \mathcal{A}, P)$ a Gaussian Hilbert space, and $\Xi \colon \mathbb{A} \to (\mathbb{U} \to \mathbb{H})$ such that $\tilde{F}(a, \,\cdot\,)(u) \in \Xi(a)(u)$ for all $a \in \mathbb{A}$ and $u \in \mathbb{U}$. Then $\Xi$ is an operator-valued Gaussian process with mean function $\mathcal{M} \colon \mathbb{A} \to \mathbb{U}$ and covariance kernel $\mathcal{K} \colon \mathbb{A} \times \mathbb{A} \to (\mathbb{U} \to \mathbb{U})$ if and only if*

*(a) $\tilde{F}$ is a $(\mathbb{U}^{\#}, \sigma(\delta_\mathbb{U}))$-valued Gaussian process with mean $\tilde{\mathcal{M}} \colon \mathbb{A} \to \mathbb{U}'$ and covariance function $\tilde{\mathcal{K}} \colon \mathbb{A} \times \mathbb{A} \to (\delta_\mathbb{U} \to \mathbb{U}')$,*

*(b)* $(u, \omega) \mapsto \tilde{F}(a, \omega)(u)$ *is mean-square continuous[9] for all $a \in \mathbb{A}$, and*

*(c)* $u \mapsto \tilde{\mathcal{K}}(a_1, a_2)(\delta_u) \in \mathbb{U}'$ *is norm-continuous for all $a_1, a_2 \in \mathbb{A}$.*

*We have $\langle \mathcal{M}(a), \cdot \rangle_{\mathbb{U}} = \tilde{\mathcal{M}}(a)$ for $a \in \mathbb{A}$, and $\langle \mathcal{K}(a_1, a_2)(u), \cdot \rangle_{\mathbb{U}} = \tilde{\mathcal{K}}(a_1, a_2)(\delta_u)$ for all $a_1, a_2 \in \mathbb{A}$ and $u \in \mathbb{U}$.*

*Proof.* This follows from Corollary A.14. $\qquad\square$

Moreover, the following results show that operator-valued Gaussian processes whose kernels map into the trace-class are "equivalent to" Gaussian processes with values in $(\mathbb{U}, \mathcal{B}(\mathbb{U}))$.

**Proposition A.19.** *Let F be a $\mathbb{U}$-valued Gaussian process on $(\Omega, \mathcal{A}, \mathrm{P})$ with index set $\mathbb{A}$, mean function $\mathcal{M}$, and covariance function $\mathcal{K}$. Define $\mathbb{H}$ as the $\mathrm{L}_2(\Omega, \mathcal{A}, \mathrm{P})$ closure of $\mathrm{span}\{\omega \mapsto \langle u, F(a, \omega) \rangle_{\mathbb{U}} \mid a \in \mathbb{A}, u \in \mathbb{U}\} \subset \mathrm{L}_2(\Omega, \mathcal{A}, \mathrm{P})$. Then*

$$\Xi \colon \mathbb{A} \to (\mathbb{U} \to \mathbb{H}), a \mapsto (u \mapsto \langle u, F(a, \cdot) \rangle_{\mathbb{U}})$$

*is an operator-valued Gaussian process with mean $\mathcal{M}$ and covariance kernel $\mathcal{K}$. Moreover, $\mathcal{K}(a, a)$ is trace-class for all $a \in \mathbb{A}$.*

*Proof.* Let $a \in \mathbb{A}$. For all $u \in \mathbb{U}$ we have

$$\|\Xi(a)(u)\|_{\mathbb{H}}^2 = \int_{\Omega} \left( \langle u, F(a, \cdot) \rangle \right)^2 \mathrm{P}(\mathrm{d}\omega) \leq \|u\|_{\mathbb{U}}^2 \underbrace{\int_{\Omega} \|F(a, \cdot)\|_{\mathbb{U}}^2 \mathrm{P}(\mathrm{d}\omega)}_{< \infty}$$

by the Cauchy-Schwarz inequality and Fernique's theorem (Bogachev, 1998, Theorem 2.8.5), i.e. $\Xi(a)$ is a bounded linear operator. $\qquad\square$

**Theorem A.20.** *Let $\Xi$ be an operator-valued Gaussian process with Gaussian Hilbert space $\mathbb{H} \subset \mathrm{L}_2(\Omega, \mathcal{A}, \mathrm{P})$, mean function $\mathcal{M}$, and covariance kernel $\mathcal{K}$ such that $\mathcal{K}(a, a)$ is trace class for all $a \in \mathbb{A}$. Then there is a $\mathbb{U}$-valued Gaussian process $F \sim \mathcal{GP}(\mathcal{M}, \mathcal{K})$ on $(\Omega, \mathcal{A}, \mathrm{P})$ such that $\langle u, F(a) \rangle_{\mathbb{U}} \in \Xi(a)(u)$ for all $a \in \mathbb{A}$ and $u \in \mathbb{U}$.*

*Proof.* First, we will construct $F \colon \mathbb{A} \times \Omega \to \mathbb{U}$. Let $a \in \mathbb{A}$. Since $\mathbb{U}$ is separable and $\mathcal{K}(a, a)$ is self-adjoint, positive-semidefinite, and trace class (and hence compact), there is an ONB $\{\psi_i^a\}_{i \in I}$ of $\mathbb{U}$ consisting of eigenvectors of $\mathcal{K}(a, a)$ (Conway, 1997, Corollary 5.4). For every $i \in I$, fix $z_i^a \in (\Xi(a)(\psi_i^a) - \langle \psi_i^a, \mathcal{M}(a) \rangle)$. We will now show that the series

$$\sum_{i \in I} z_i^a \psi_i^a \qquad\qquad (A.3)$$

converges P-almost surely in $\mathbb{U}$. The $\{z_i^a\}_{i \in I}$ are centered, independent Gaussian random variables with variances given by the eigenvalues $\lambda_i^a$ of $\mathcal{K}(a, a)$, since

$$\mathbb{E}_{\mathrm{P}}[z_i^a] = \mathbb{E}_{\mathrm{P}}[\Xi(a)(\psi_i^a)] - \langle \psi_i^a, \mathcal{M}(a) \rangle = 0$$

and

$$\mathrm{Cov}_{\mathrm{P}}[z_i^a, z_j^a] = \mathrm{Cov}_{\mathrm{P}}[\Xi(a)(\psi_i^a), \Xi(a)(\psi_j^a)] = \langle \psi_i^a, \mathcal{K}(a, a)(\psi_j^a) \rangle_{\mathbb{U}} = \langle \psi_i^a, \lambda_j^a \psi_j^a \rangle_{\mathbb{U}} = \lambda_j^a \delta_{ij}.$$

We have $\sum_{i \in I} \lambda_i^a < \infty$ because the operator $\mathcal{K}(a, a)$ is trace-class. By Theorem 1.1.4 in Bogachev (1998), this means that

$$\sum_{i \in I} z_i^a = \sum_{i \in I} z_i^a \|\psi_i^a\|_{\mathbb{U}}$$

converges P-almost surely, and hence Equation (A.3) P-almost surely absolutely convergent in $\mathbb{U}$. Define

$$F(a) \stackrel{\text{a.s.}}{:=} \mathcal{M}(a) + \sum_{i \in I} z_i^a \psi_i^a.$$

---

[9] $\tilde{F}(a, \cdot)(u) \xrightarrow{\mathrm{L}_2} \tilde{F}(a, \cdot)(u_0)$ as $u \xrightarrow{\mathbb{U}} u_0$

Then $\langle u, \mathrm{F}(a) \rangle \in \Xi(a)(u)$ for $u \in \mathrm{span}\{\psi_i^a\}_{i \in I}$.

We will now show that $\langle u, \mathrm{F}(a) \rangle \in \Xi(a)(u)$ for all $a \in \mathbb{A}$ and $u \in \mathbb{U}$. To this end, fix $u \in \mathbb{U}$ and $\mathrm{y}_{a,u} \in \Xi(a)(u)$. There is $\{u_n\}_{n=1}^\infty \subset \mathrm{span}\{\psi_i^a\}_{i \in I}$ such that $u = \lim_{n \to \infty} u_n$. Boundedness of $\Xi(a)$ implies $\langle u_n, \mathrm{F}(a) \rangle \xrightarrow{\mathrm{L}^2} \mathrm{y}_{a,u}$ and, by Remark 6.11 and Corollary 6.13 in Klenke (2014), there exists a subsequence $\{u_{n_k}\}_{k=1}^\infty \subset \{u_n\}_{n=1}^\infty$ such that $\langle u_{n_k}, \mathrm{F}(a) \rangle \xrightarrow{\mathrm{a.s.}} \mathrm{y}_{a,u}$. Moreover, by continuity of the inner product, we know that $\langle u_{n_k}, \mathrm{F}(a, \omega) \rangle \to \langle u, \mathrm{F}(a, \omega) \rangle$ for all $\omega \in \Omega$, and hence $\langle u_{n_k}, \mathrm{F}(a) \rangle \xrightarrow{\mathrm{a.s.}} \langle u, \mathrm{F}(a) \rangle$. Since almost sure limits are unique up to almost sure equality (Klenke, 2014), we have $\mathrm{y}_{a,u} \stackrel{\mathrm{a.s.}}{=} \langle u, \mathrm{F}(a) \rangle$. Hence, $\langle u, \mathrm{F}(a) \rangle \in \Xi(a)(u)$.

The rest of the claim now follows from Corollary A.13 with $\mathrm{f}((a, u), \omega) \coloneqq \langle u, \mathrm{F}(a, \omega) \rangle$ and $\hat{\mathbb{L}} = \mathbb{U}'$. $\qquad\square$

# B. Linearized Laplace Approximation

The *linearized Laplace approximation* (LLA) (MacKay, 1992a;b; Immer et al., 2021) is a conceptually simple, yet effective (Daxberger et al., 2021a) method for obtaining an approximate posterior distribution over the parameters $\boldsymbol{w} \in \mathbb{R}^p$ of a neural network $\boldsymbol{f} \colon \mathbb{R}^d \times \mathbb{R}^p \to \mathbb{R}^{d'}$. It applies whenever the objective function $R$ used to train the neural network is (equivalent to) a negative log-posterior

$$R(\boldsymbol{w}) = -\log p(\boldsymbol{w} \mid \mathcal{D}) = -\log p(\boldsymbol{w}) - \sum_{i=1}^n \log p(\boldsymbol{y}^{(i)} \mid \boldsymbol{f}(\boldsymbol{x}^{(i)}, \boldsymbol{w})) + \mathrm{const.}$$

of the network parameters given data $\mathcal{D} = \{(\boldsymbol{x}^{(i)}, \boldsymbol{y}^{(i)})\}_{i=1}^n$. It is common for the prior over the parameters to be Gaussian, in which case $-\log p(\boldsymbol{w})$ acts as an L2-regularizer on the parameters. During training, we attempt to find a local minimum $\boldsymbol{w}^\star$ of the objective function $R$, i.e. a maximum a-posteriori (MAP) estimator of the network parameters given the data.

Following Immer et al. (2021), we approximate the posterior of the network weights as follows: First, we linearize the model using a first-order Taylor approximation in the weights around the MAP estimator $\boldsymbol{w}^\star$

$$\boldsymbol{f}(\boldsymbol{x}, \boldsymbol{w}) \approx \boldsymbol{f}_{\boldsymbol{w}^\star}^{\mathrm{lin}}(\boldsymbol{x}, \boldsymbol{w}) \coloneqq \boldsymbol{f}(\boldsymbol{x}, \boldsymbol{w}^\star) + \mathrm{D}_{\boldsymbol{w}} \boldsymbol{f}(\boldsymbol{x}, \boldsymbol{w})|_{\boldsymbol{w}^\star} (\boldsymbol{w} - \boldsymbol{w}^\star).$$

Afterwards, we compute a second-order Taylor approximation of the negative log-posterior $R_{\boldsymbol{w}^\star}^{\mathrm{lin}}$ of the linearized network at the MAP $\boldsymbol{w}^\star$

$$
\begin{aligned}
R_{\boldsymbol{w}^\star}^{\mathrm{lin}}(\boldsymbol{w}) &\coloneqq -\log p(\boldsymbol{w}) - \sum_{i=1}^n \log p(\boldsymbol{y}^{(i)} \mid \boldsymbol{f}_{\boldsymbol{w}^\star}^{\mathrm{lin}}(\boldsymbol{x}^{(i)}, \boldsymbol{w})) + \mathrm{const.} \\
&\approx R(\boldsymbol{w}^\star) + \underbrace{\nabla R(\boldsymbol{w}^\star)^\top}_{\approx 0}(\boldsymbol{w} - \boldsymbol{w}^\star) + \frac{1}{2}(\boldsymbol{w} - \boldsymbol{w}^\star)^\top \boldsymbol{P}(\boldsymbol{w} - \boldsymbol{w}^\star) \\
&= \frac{1}{2}(\boldsymbol{w} - \boldsymbol{w}^\star)^\top \boldsymbol{P}(\boldsymbol{w} - \boldsymbol{w}^\star) + \mathrm{const.}
\end{aligned}
\tag{B.1}
$$

with $\boldsymbol{P} \coloneqq -\mathrm{H}_{\boldsymbol{w}} \log p(\boldsymbol{w})|_{\boldsymbol{w}^\star} + \boldsymbol{G}$, where

$$\boldsymbol{G} \coloneqq -\sum_{i=1}^n \mathrm{D}_{\boldsymbol{w}} \boldsymbol{f}(\boldsymbol{x}^{(i)}, \boldsymbol{w})\Big|_{\boldsymbol{w}^\star} \mathrm{H}_{\boldsymbol{f}} \log p(\boldsymbol{y}^{(i)} \mid \boldsymbol{f})\Big|_{\boldsymbol{f}(\boldsymbol{x}^{(i)}, \boldsymbol{w}^\star)} \mathrm{D}_{\boldsymbol{w}} \boldsymbol{f}(\boldsymbol{x}^{(i)}, \boldsymbol{w})\Big|_{\boldsymbol{w}^\star}^\top \tag{B.2}$$

is the so-called *generalized Gauss-Newton* (GGN) matrix (Schraudolph, 2002). The GGN is guaranteed to be positive-semidefinite. Equation (B.1) is the negative log-density of a (potentially degenerate) multivariate Gaussian distribution with mean $\boldsymbol{w}^\star$ and covariance matrix $\boldsymbol{P}^\dagger$, i.e.

$$p(\mathbf{w} = \boldsymbol{w} \mid \mathcal{D}) = \exp R(\boldsymbol{w}) \approx \exp R_{\boldsymbol{w}^\star}^{\mathrm{lin}}(\boldsymbol{w}) \approx \mathcal{N}\left(\boldsymbol{w}; \boldsymbol{w}^\star, \boldsymbol{P}^\dagger\right).$$

This Gaussian distribution is referred to as the *linearized Laplace approximation* of $p(\mathbf{w} = \boldsymbol{w} \mid \mathcal{D})$. Under the linearized model, the approximate Gaussian posterior over the weights induces a tractable posterior predictive over the output of the neural network (Khan et al., 2019; Immer et al., 2021). More precisely, using closure properties of Gaussian distributions under affine maps, one can show that the pushforward of the LLA posterior through $\boldsymbol{w} \mapsto \boldsymbol{f}_{\boldsymbol{w}^\star}^{\mathrm{lin}}(\boldsymbol{x}, \boldsymbol{w})$ defines a ($d'$-output) Gaussian process

$$\mathbf{f} \mid \mathcal{D} \sim \mathcal{GP}\left(\boldsymbol{f}(\cdot, \boldsymbol{w}^\star), (\boldsymbol{x}_1, \boldsymbol{x}_2) \mapsto \mathrm{D}_{\boldsymbol{w}} \boldsymbol{f}(\boldsymbol{x}_1, \boldsymbol{w})|_{\boldsymbol{w}^\star} \boldsymbol{P}^\dagger \mathrm{D}_{\boldsymbol{w}} \boldsymbol{f}(\boldsymbol{x}_2, \boldsymbol{w})|_{\boldsymbol{w}^\star}^\top\right).$$

# C. Implementation Details

## C.1. Last-Layer LUNO for FNOs

For an input $\boldsymbol{a} \in \mathbb{A}$, we can factorize the FNO as

$$\boldsymbol{F}(\boldsymbol{a}, \boldsymbol{w})(\boldsymbol{x}) = \underbrace{(\boldsymbol{q}(\cdot, \boldsymbol{w}_q) \circ \sigma^{(L-1)})}_{=:\tilde{\boldsymbol{q}}} \left( \boldsymbol{z}^{(L-1)}(\boldsymbol{x}, \boldsymbol{w}_{L-1}) \right),$$

with

$$
\begin{aligned}
\boldsymbol{z}_i^{(L-1)}(\boldsymbol{x}, \boldsymbol{w}_{L-1}) &:= \sum_{j=1}^{d'_v} \mathcal{F}^{-1}\left( \boldsymbol{R}_{:ij}^{(L-1)} \odot \hat{\boldsymbol{v}}_{:j}^{(L-1)} \right)(\boldsymbol{x}) + \sum_{j=1}^{d'_v} \boldsymbol{W}_{ij}^{(L-1)} \boldsymbol{v}_j^{(L-1)}(\boldsymbol{x}) \\
&= \sum_{j=1}^{d'_v} \sum_{k=1}^{k_{\max}} \mathrm{Re}(\boldsymbol{R}_{kij}^{(L-1)}) \underbrace{\mathrm{Re}(\hat{\boldsymbol{v}}_{kj}^{(L-1)}) \cos\left(\langle \omega_k, \boldsymbol{x} \rangle\right)}_{=:\phi_{kj}(\boldsymbol{x})} \\
&\quad + \sum_{j=1}^{d'_v} \sum_{k=1}^{k_{\max}} \mathrm{Im}(\boldsymbol{R}_{kij}^{(L-1)}) \underbrace{(-1)\,\mathrm{Im}(\hat{\boldsymbol{v}}_{kj}^{(L-1)}) \sin\left(\langle \omega_k, \boldsymbol{x} \rangle\right)}_{=:\varphi_{kj}(\boldsymbol{x})} \\
&\quad + \sum_{j=1}^{d'_v} \boldsymbol{W}_{ij}^{(L-1)} \underbrace{\boldsymbol{v}_j^{(L-1)}(\boldsymbol{x})}_{=:\psi_j(\boldsymbol{x})},
\end{aligned}
$$

where $\hat{\boldsymbol{v}}_{kj}^{(L-1)} := \mathcal{F}(\boldsymbol{v}_j^{(L-1)})_k \in \mathbb{C}$ for $k \in \{1, \dots, k_{\max}\}$. We note that $\boldsymbol{z}^{(L-1)}(\boldsymbol{x}, \boldsymbol{w}_{L-1})$ is linear in $\boldsymbol{w}_{L-1}$. Thus, assuming a Gaussian belief over

$$\boldsymbol{w}_{L-1} \cong \left( \mathrm{Re}(\boldsymbol{R}^{(L-1)}), \mathrm{Im}(\boldsymbol{R}^{(L-1)}), \boldsymbol{W}^{(L-1)} \right)$$

induces a (multi-output) GP over $\boldsymbol{z}^{(L-1)}$:

$$
\begin{aligned}
\mathbf{z}^{(L-1)} &\sim \mathcal{GP}\left( \boldsymbol{m}_{\mathbf{z}^{(L-1)}}, \boldsymbol{K}_{\mathbf{z}^{(L-1)}} \right) \qquad \text{with} \\
\boldsymbol{m}_{\mathbf{z}^{(L-1)}} &= \boldsymbol{z}^{(L-1)}(\boldsymbol{x}, \boldsymbol{w}_{L-1}^\star).
\end{aligned}
$$

Moreover, $\mathbf{z}^{(L-1)}$ is the sum of three (dependent) parametric Gaussian processes with feature functions $\phi_{kj}$, $\varphi_{kj}$, and $\psi_j$, respectively. Consequently, the function-valued GP induced by the linearized FNO is given by

$$\mathrm{F}(\boldsymbol{a})(\boldsymbol{x}) = \tilde{\boldsymbol{q}}(\boldsymbol{m}_{\mathbf{z}^{(L-1)}}(\boldsymbol{x})) + \mathrm{D}\tilde{\boldsymbol{q}}\left( \boldsymbol{m}_{\mathbf{z}^{(L-1)}}(\boldsymbol{x}) \right) (\mathbf{z}^{(L-1)}(\boldsymbol{x}) - \boldsymbol{m}_{\mathbf{z}^{(L-1)}}(\boldsymbol{x})),$$

i.e. $\mathrm{F}(\boldsymbol{a}) \sim \mathcal{GP}\left( \boldsymbol{m}_{\boldsymbol{a}}, \boldsymbol{K}_{\boldsymbol{a}} \right)$ with

$$
\begin{aligned}
\boldsymbol{m}_{\boldsymbol{a}}(\boldsymbol{x}) &= \boldsymbol{F}(\boldsymbol{a}, \boldsymbol{w}^\star)(\boldsymbol{x}), \quad \text{and} \\
\boldsymbol{K}_{\boldsymbol{a}}(\boldsymbol{x}_1, \boldsymbol{x}_2) &= \mathrm{D}\tilde{\boldsymbol{q}}\left( \boldsymbol{m}_{\mathbf{z}^{(L-1)}}(\boldsymbol{x}_1) \right) \boldsymbol{K}_{\mathbf{z}^{(L-1)}}(\boldsymbol{x}_1, \boldsymbol{x}_2) \mathrm{D}\tilde{\boldsymbol{q}}\left( \boldsymbol{m}_{\mathbf{z}^{(L-1)}}(\boldsymbol{x}_2) \right)^\top.
\end{aligned}
$$

If the input function $\boldsymbol{a} \in \mathbb{A}$ is discretized on a grid $\boldsymbol{X}_{\mathbb{A}}^{(i)} \in (\mathbb{D}_{\mathbb{A}})^{n_{\mathbb{A}}^{(i)}}$, we set $\hat{\boldsymbol{v}}_{kj}^{(L-1)} := \mathrm{rfft}(\boldsymbol{v}_j^{(L-1)}(\boldsymbol{X}_{\mathbb{A}}^{(i)}))_k \in \mathbb{C}$ and $\psi_j(\boldsymbol{x})$ interpolates $\boldsymbol{v}_j^{(L-1)}(\boldsymbol{X}_{\mathbb{A}}^{(i)})$ (e.g. spline interpolation or Fourier interpolation).

# D. Experimental details

## D.1. Data: PDE trajectories

### D.1.1. LOW DATA GENERATION USING APEBENCH

To evaluate the performance of the uncertainty quantification methods discussed, we utilize the code in the APEBench for generating data from Burgers', Hyper Diffusion and Kuramoto-Sivashinsky equation (conservative) (cf. (Koehler et al., 2024) for more details). Table 3 summarizes the characteristics of the datasets we use, the number of trajectories for training, and testing, as well as the spatial and temporal resolutions.

| PDE Name | Dimensions | Training Traj. | Valid. Traj. | Test Traj. | Spatial Res. | Temp. Res. |
| --- | --- | --- | --- | --- | --- | --- |
| Burgers | 1D | 25 | 250 | 250 | 256 | 59 |
| Hyper Diffusion | 1D | 25 | 250 | 250 | 256 | 59 |
| Kuramoto-Sivashinsky (cons.) | 1D | 25 | 250 | 250 | 256 | 59 |

Table 3: Summary of PDE datasets generated using APEBench.

### D.1.2. OUT-OF-DISTRIBUTION DATA GENERATION

We generated additional datasets based on the advection-diffusion-reaction equation

$$\frac{\partial u}{\partial t} + \mathbf{v} \cdot \nabla u = \alpha \nabla^2 u + R, \tag{D.1}$$

to evaluate the robustness of the uncertainty quantification methods under out-of-distribution (OOD) scenarios. Here $u$ represents the scalar field (e.g., concentration or temperature), $\mathbf{v}$ is the velocity field, $\alpha$ is the diffusion coefficient, and $R$ is the reaction term, where $\alpha = 0.026$ was held constant throughout the datasets. The following specifications guided this process:

- **Datasets:** Trajectories for five variations of the advection-diffusion-reaction equation were generated (cf. Figure 5):
  1. Base
     - Random number (1-10) of Gaussian blobs as an initial condition.
     - Constant random velocity field.
     - No reaction terms.
  2. Flip
     - Random number (1-10) of Gaussian blobs as an initial condition.
     - Constant random velocity field, which is reversed at the center of the domain.
     - No reaction terms.
  3. Pos
     - Random number (1-10) of Gaussian blobs as an initial condition.
     - Constant random velocity field.
     - Randomly placed triangular heat source.
  4. Pos-Neg
     - Random number (1-10) of Gaussian blobs as an initial condition.
     - Constant random velocity field.
     - Randomly placed triangular heat source; randomly placed cloud-shaped heat sink.
  5. Pos-Neg-Flip
     - Random number (1-10) of Gaussian blobs as an initial condition.
     - Constant random velocity field, which is reversed at the center of the domain.
     - Randomly placed triangular heat source; randomly placed cloud-shaped heat sink.

- **PDE Solver:** We utilized a custom implementation to solve the advection-diffusion-reaction equation, relying on a 9-point stencil with Runge-Kutta 4 (RK4) with fine spatial and temporal resolutions.

- **Simulation Parameters:**
    - Spatial resolution: $100 \times 100$ grid.
    - Temporal resolution: $\Delta t = 5 \times 10^{-10}$ over 200-time steps, from which 59 are sub-sampled.

We train a Fourier neural operator on 1000 training trajectories of the Base dataset and calibrate uncertainty quantification methods - if necessary - on a corresponding validation dataset with 250 input-output pairs. For the evaluation, we consider 250 random test pairs from some of the datasets. In Figure 5, we depict a single trajectory for each of these 5 datasets generated. To seamlessly evaluate the various OOD datasets, we pad the training set with constant zeros for the velocity field and reaction terms as a placeholder.

## D.2. Model and training

For all experiments, we consider the original Fourier neural operator architecture (Li et al., 2021) with the hyperparameter suggestions following (Koehler et al., 2024), i.e. 12 modes (per spatial dimension) and 18 hidden dimensions constant throughout the network, with a total of 4 Fourier blocks. For the training, we consider 10 initial time steps and train to predict the following time step. The velocity field and the reaction term are glued to the input. Furthermore, the input is always padded by two constant zero grid points to reduce artifacts at the borders. Networks for the low data experiment are trained for 100 epochs, all remaining networks are trained for 1000 epochs—where one epoch corresponds to iterating through a single input-output pair per trajectory in the training set. During training the mean squared error loss was minimized using AdamW (Loshchilov & Hutter, 2019) combined with a cosine decay learning rate scheduler with warmup. All training implementations rely on jax (Bradbury et al., 2018), Flax NNX and optax.

## D.3. Uncertainty quantification methods

### D.3.1. INPUT PERTURBATIONS

Input Perturbations involve augmenting input data with small, random perturbations to introduce diversity into the model's predictions. This approach exploits the model's sensitivity to input variations to approximate uncertainty in the output space. Following the approach of Pathak et al. (2022), ensemble predictions are generated by forwarding a batch of pointwise perturbed versions of a single input $u_n$. Perturbations are sampled as $\epsilon_{x,t} \sim \mathcal{N}(0, \sigma^2)$ for each input value of the discretized function state $u_n(x, t)$. The parameter $\sigma$ is calibrated to achieve accurate marginal uncertainty predictions.

### D.3.2. ENSEMBLES

Deep ensembles rely on training multiple independent instances of a model, each initialized with different random seeds and potentially trained on different subsets of the data (Lakshminarayanan et al., 2017). The diversity in the learned weight configurations leads to a variety of predictions for a given input, enabling the computation of both mean predictions and marginal uncertainty estimates. From a Bayesian perspective these weights represent close to true samples from the weight space posterior distribution, that can only be found through the high cost of additional training runs.

### D.3.3. ISOTROPIC GAUSSIAN (∗-ISO)

The isotropic Gaussian covariance structure represents the weight space uncertainty as:

$$\mathcal{N}(\boldsymbol{w}^\star, \boldsymbol{\Sigma} := \sigma^2 \mathbf{I}) \tag{D.2}$$

where $\sigma^2$ is the variance parameter and $\mathbf{I}$ is the identity matrix, reflecting independence and identical uncertainty across all dimensions in the weight space. To calibrate the uncertainty, we tune the parameter $\sigma^2$ as outlined below.

From a Bayesian perspective, as seen in methods such as Laplace, this can be viewed as just considering a calibrated prior over the selected weight space.

### D.3.4. LAPLACE APPROXIMATION (∗-LA)

In the linearized setting, a natural choice for a posterior Gaussian belief in weight space is given by the linearized Laplace approximation, where the Hessian is given by the Generalized Gauss-Newton (GGN) $\mathbf{H}_{\mathrm{GGN}}$ (Immer et al., 2021). For further details, see Equation (B.1).

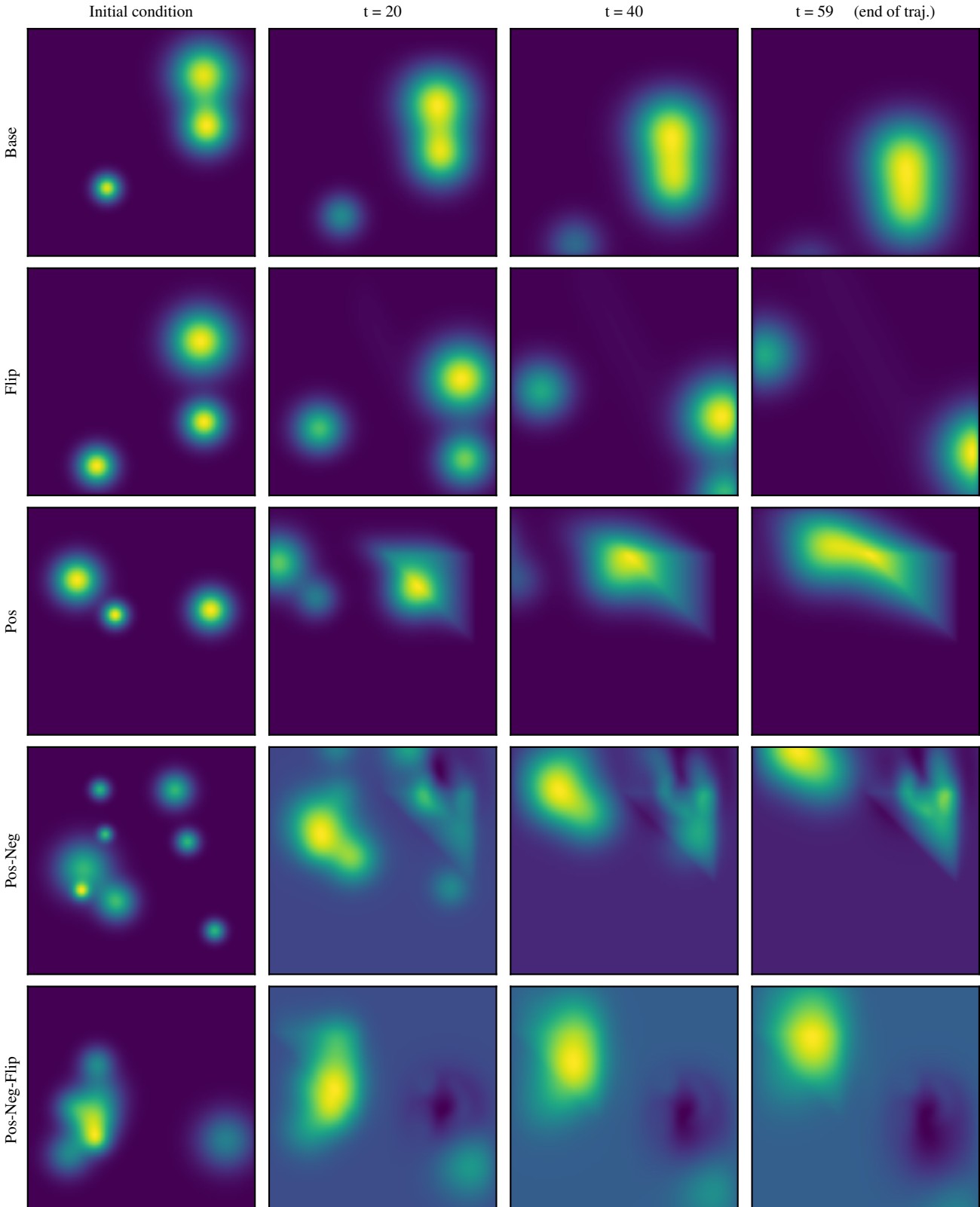

Figure 5: Initial condition and three-time steps of a single trajectory per generated dataset (Base, Flip, Pos, Pos-Neg, Pos-Neg-Flip).

For high-dimensional parameter spaces, approximations of the GGN are required. Common techniques include diagonal approximations (Daxberger et al., 2021a), K-FAC (Martens & Grosse, 2015), or low-rank approximations (Dangel et al., 2022). Since diagonal and K-FAC approximations do not capture correlations between weights acting on different Fourier modes, we focus on a low-rank approximation of the GGN instead.

We extend the approach in (Dangel et al., 2022) by selecting the largest eigenspaces of the GGN and placing an isotropic Gaussian prior over all weights instead of approximating the posterior covariance directly. This allows regions of uncertainty to fall back to the prior belief. For a fixed input function and evaluation grid, the push-forward with the low-rank approximation $\mathbf{H}_{\text{GGN}} = VV^T$ yields the normal distribution:

$$\mathcal{N}(\mathcal{F}(\mathbf{x}_\cdot, \boldsymbol{\theta}), \mathbf{J}_{\boldsymbol{\theta}} \boldsymbol{\Sigma} \mathbf{J}_{\boldsymbol{\theta}}^T), \tag{D.3}$$

where $\boldsymbol{\Sigma} = (nVV^T + \sigma\mathbf{I})^{-1}$, $n$ is the number of input-output pairs used to train the neural operator, and $\mathbf{J}_{\boldsymbol{\theta}}$ is the Jacobian of the model with respect to the weights. In all experiments, we consider a low rank of 500. For the low data regime, we consider all input-output pairs, while for the OOD experiment only a minibatch of 1000 input-output pairs.

### D.3.5. SAMPLE-∗

For each weight-space covariance method and the input perturbation approach, we consider a sample-based pushforward which generates an ensemble of predictions in the output space. This aligns with the way most weight-space covariance methods are introduced in the literature (cf. Maddox et al., 2019). In our experiments, we generate 200 samples that are propagated through the network. The empirical mean and standard deviation are then estimated using the set of predictions in the output space.

### D.3.6. LUNO-∗

The core implementation of LUNO leverages matrix-free Jacobian-vector products and the algebraic structure of the chosen Gaussian covariance matrix in weight space. Our framework supports fully lazy evaluations, enabling efficient sampling, marginal variance estimation, and matrix-vector products with the covariance of the output space.

When restricting the weight space uncertainty to only the last Fourier block, we can explicitly derive the matrix action of the Jacobian of the inverse Fast Fourier Transform (IFFT). Additionally, we exploit the fact that the final linear layer is applied pointwise in the spatial domain, significantly improving computational efficiency and reducing memory requirements compared to traditional implementations of linearized pushforwards (Daxberger et al., 2021a).

## D.4. Evaluation

We use the following metrics to evaluate model performance on 250 input-output pairs from the respective test trajectories. Here, $y_i$ denotes the ground truth, $\hat{y}_i$ represents the predicted mean, and $\sigma_i$ is the predicted standard deviation for the $i$-th sample. All reported numbers are the expected value over the test samples.

### D.4.1. ROOT MEAN SQUARED ERROR (RMSE)

The root mean squared error

$$\text{RMSE} = \sqrt{\frac{1}{n}\sum_{i=1}^{n}(y_i - \hat{y}_i)^2}$$

measures the average magnitude of the errors between ensemble mean/linearized mean and ground truth values. Lower RMSE indicates better predictive accuracy, with zero being the ideal value.

### D.4.2. MARGINAL NEGATIVE LOG-LIKELIHOOD (NLL)

The marginal NLL quantifies how well the predictive distribution fits the data under the assumption of Gaussian uncertainty. Lower NLL values indicate better calibration of the uncertainty estimates and higher likelihood of the observed data under the predictive model. It represents a trade-off between lower variances and how much of the error is accurately captured by the uncertainty.

$$\text{NLL} = -\sum_{i=1}^{n}\log\left(\frac{1}{\sqrt{2\pi\sigma_i^2}}\exp\left(-\frac{(y_i - \hat{y}_i)^2}{2\sigma_i^2}\right)\right)$$

### D.4.3. $\chi^2$-STATISTIC

The $\chi^2$-statistic measures the average squared error normalized by the predicted variance. A value close to 1 indicates well-calibrated uncertainty predictions.

$$Q = \frac{1}{n} \sum_{i=1}^{n} \frac{(y_i - \hat{y}_i)^2}{\sigma_i^2}$$

Values above one suggest overconfidence, while values below one indicate underconfident predictive uncertainty. From a UQ perspective underconfidence is better than overconfidence.

## D.5. Calibration

All hyperparameters of the discussed UQ methods (mostly $\sigma^2$ in the above definitions) were calibrated using 250 input-output pairs of the validation set to minimize the marginal negative log-likelihood. We calibrate each method's hyperparameters separately using grid search over a logarithmically spaced grid with 500 points centered around the relevant value.

## D.6. Additional results

### D.6.1. LOW DATA REGIME

| Method | RMSE ($\downarrow$) | $\chi^2$ | NLL ($\downarrow$) |
|---|---|---|---|
| Input Perturbations | $1.98 \times 10^{-2}$ | 1.203 | $-2.3927$ |
| Ensemble | $\mathbf{1.95 \times 10^{-2}}$ | 12.674 | 1.8957 |
| Sample-Iso | $2.02 \times 10^{-2}$ | 1.237 | $-2.4391$ |
| LUNO-Iso | $1.99 \times 10^{-2}$ | 1.155 | $-2.4677$ |
| Sample-LA | $2.18 \times 10^{-2}$ | 2.097 | $-2.2312$ |
| LUNO-LA | $1.99 \times 10^{-2}$ | 0.984 | $\mathbf{-2.5248}$ |

Table 4: Performance metrics comparison for UQ methods evaluated on an FNO trained on 25 trajectories of the Hyper-Diffusion equation.

| Method | RMSE ($\downarrow$) | $\chi^2$ | NLL ($\downarrow$) |
|---|---|---|---|
| Input Perturbations | $7.84 \times 10^{-2}$ | 0.938 | $-1.0385$ |
| Ensembles | $\mathbf{6.82 \times 10^{-2}}$ | 7.618 | 0.8489 |
| Sample-Iso | $7.88 \times 10^{-2}$ | 1.211 | $-1.0862$ |
| LUNO-Iso | $7.82 \times 10^{-2}$ | 0.922 | $-1.0995$ |
| Sample-LA | $1.46 \times 10^{-1}$ | 2.758 | $-0.1699$ |
| LUNO-LA | $7.82 \times 10^{-2}$ | 1.058 | $\mathbf{-1.1653}$ |

Table 5: Performance metrics comparison for UQ methods evaluated on an FNO trained on 25 trajectories of the Kuramoto-Sivashinsky (conservative) equation.

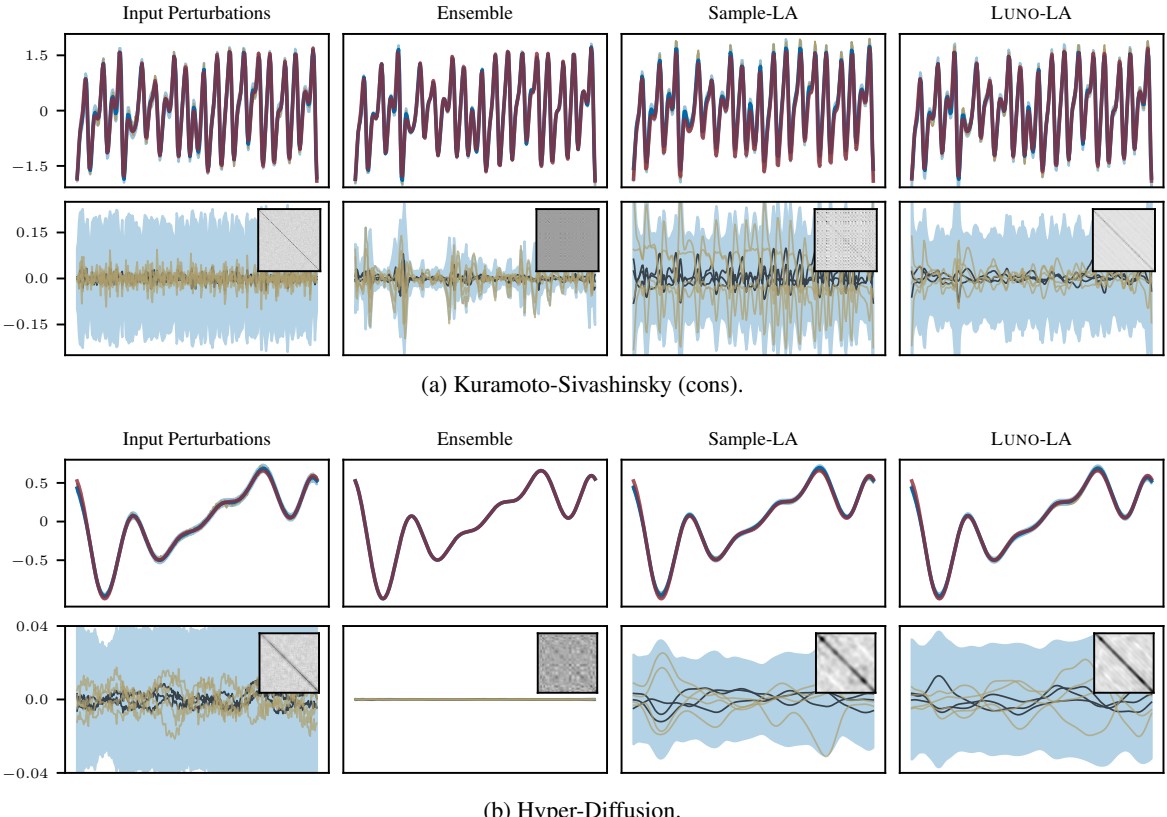

(a) Kuramoto-Sivashinsky (cons).

(b) Hyper-Diffusion.

Figure 6: FNO predictive uncertainty quantified by several different methods. Top row: target function (—), mean (—) and 1.96 standard deviations (■) of, as well as samples (—) from, the predictive belief. For the ensemble, the samples are four of the ensemble members. Bottom row: spread of the predictive distribution around the mean. For the sample-/ensemble-based methods, we construct a Gaussian distribution from the empirical covariance matrix and draw four samples (—). We plot 1.96 standard deviations (■) of the predictive belief, as well as the top-three eigenfunctions (—) and a heatmap of the predictive covariance matrix (top right corner of panels).

| Method | Base | Flip | Pos | Pos-Neg | Pos-Neg-Flip |
|---|---|---|---|---|---|
| Input Perturbations | $-2.586$ | $2.573$ | $-1.174$ | $69.346$ | $494.935$ |
| Ensemble | $\mathbf{-5.313}$ | $\mathbf{-3.825}$ | $\mathbf{-4.802}$ | $\mathbf{-1.257}$ | $\mathbf{-1.014}$ |
| Sample-Iso | $-2.921$ | $4.071$ | $-1.226$ | $9.457$ | $43.362$ |
| LUNO-Iso | $-2.892$ | $3.450$ | $-1.260$ | $7.636$ | $37.733$ |
| Sample-LA | $-2.576$ | $4.395$ | $-1.183$ | $7.369$ | $27.046$ |
| LUNO-LA | $-2.934$ | $-1.126$ | $-1.742$ | $-0.818$ | $1.164$ |

Table 6: Negative Log-Likelihood (NLL) across different OOD datasets. Lower is better.

### D.6.2. OUT-OF-DISTRIBUTION

| Method | RMSE ($\downarrow$) | $\chi^2$ | NLL ($\downarrow$) |
|---|---|---|---|
| Input Perturbations | $1.46 \times 10^{-2}$ | $1.497$ | $-2.5861$ |
| Ensemble | $\mathbf{1.51 \times 10^{-3}}$ | $0.162$ | $\mathbf{-5.3134}$ |
| Sample-Iso | $1.44 \times 10^{-2}$ | $1.217$ | $-2.9214$ |
| LUNO-Iso | $1.46 \times 10^{-2}$ | $1.189$ | $-2.8919$ |
| Sample-Iso | $1.73 \times 10^{-2}$ | $1.910$ | $-2.5756$ |
| LUNO-LA | $1.46 \times 10^{-2}$ | $0.841$ | $-2.9340$ |

Table 7: Metrics evaluated on OOD dataset Base.

| Method | RMSE ($\downarrow$) | $\chi^2$ | NLL ($\downarrow$) |
|---|---|---|---|
| Input Perturbations | $4.36 \times 10^{-2}$ | $12.255$ | $2.5727$ |
| Ensemble-Sample | $\mathbf{4.43 \times 10^{-3}}$ | $0.215$ | $\mathbf{-3.8249}$ |
| Sample-Iso | $4.37 \times 10^{-2}$ | $15.321$ | $4.0715$ |
| LUNO-Iso | $4.36 \times 10^{-2}$ | $13.983$ | $3.4502$ |
| Sample-LA | $4.62 \times 10^{-2}$ | $15.868$ | $4.3953$ |
| LUNO-LA | $4.36 \times 10^{-2}$ | $3.475$ | $-1.1257$ |

Table 8: Metrics evaluated on OOD dataset Flip.

| Method | RMSE ($\downarrow$) | $\chi^2$ | NLL ($\downarrow$) |
|---|---|---|---|
| Input Perturbations | $2.94 \times 10^{-2}$ | $4.292$ | $-1.1744$ |
| Ensemble | $\mathbf{4.93 \times 10^{-3}}$ | $0.549$ | $\mathbf{-4.8023}$ |
| Sample-Iso | $2.91 \times 10^{-2}$ | $4.552$ | $-1.2261$ |
| LUNO-Iso | $2.94 \times 10^{-2}$ | $4.408$ | $-1.2596$ |
| Sample-LA | $2.79 \times 10^{-2}$ | $4.622$ | $-1.1835$ |
| LUNO-LA | $2.94 \times 10^{-2}$ | $3.001$ | $-1.7416$ |

Table 9: Metrics evaluated on OOD dataset Pos.

| Method | RMSE ($\downarrow$) | $\chi^2$ | NLL ($\downarrow$) |
|---|---|---|---|
| Input Perturbations | $5.55 \times 10^{-2}$ | 145.561 | 69.3458 |
| Ensemble | $9.79 \times 10^{-2}$ | 1.161 | $\mathbf{-1.2569}$ |
| Sample-Iso | $5.55 \times 10^{-2}$ | 25.961 | 9.4566 |
| LUNO-Iso | $\mathbf{5.52 \times 10^{-2}}$ | 22.231 | 7.6355 |
| Sample-LA | $5.94 \times 10^{-2}$ | 21.685 | 7.3688 |
| LUNO-LA | $\mathbf{5.52 \times 10^{-2}}$ | 4.182 | $-0.8180$ |

Table 10: Metrics evaluated on OOD dataset Pos-Neg.

| Method | RMSE ($\downarrow$) | $\chi^2$ | NLL ($\downarrow$) |
|---|---|---|---|
| Input Perturbations | $\mathbf{1.10 \times 10^{-1}}$ | 997.143 | 494.9350 |
| Ensemble | $1.39 \times 10^{-1}$ | 1.006 | $\mathbf{-1.0140}$ |
| Sample-Iso | $\mathbf{1.10 \times 10^{-1}}$ | 93.712 | 43.3620 |
| Prior-Iso | $\mathbf{1.10 \times 10^{-1}}$ | 82.367 | 37.7331 |
| Sample-LA | $1.15 \times 10^{-1}$ | 60.728 | 27.0460 |
| LUNO-LA | $\mathbf{1.10 \times 10^{-1}}$ | 7.127 | 1.1642 |

Table 11: Metrics evaluated on OOD dataset Pos-Neg-Flip.

### D.6.3. EVALUATION OF A SINGLE TRAJECTORY

**Run-Time comparison**

| Method | LUNO-$*$ | Sample-$*$ |
|---|---|---|
| Input Perturbations | – | $10.19 \pm 0.006$ |
| Ensemble | – | $0.85 \pm 0.004$ |
| $*$-Iso | $0.53 \pm 0.004$ | $14.00 \pm 0.008$ |
| $*$-LA | $5.75 \pm 0.017$ | $27.70 \pm 0.006$ |

Table 12: Comparison of run-time performance between sampling-based (Sample-$*$) and linearization-based (LUNO-$*$) methods across different uncertainty quantification techniques when out-rolling a single trajectory iteratively.

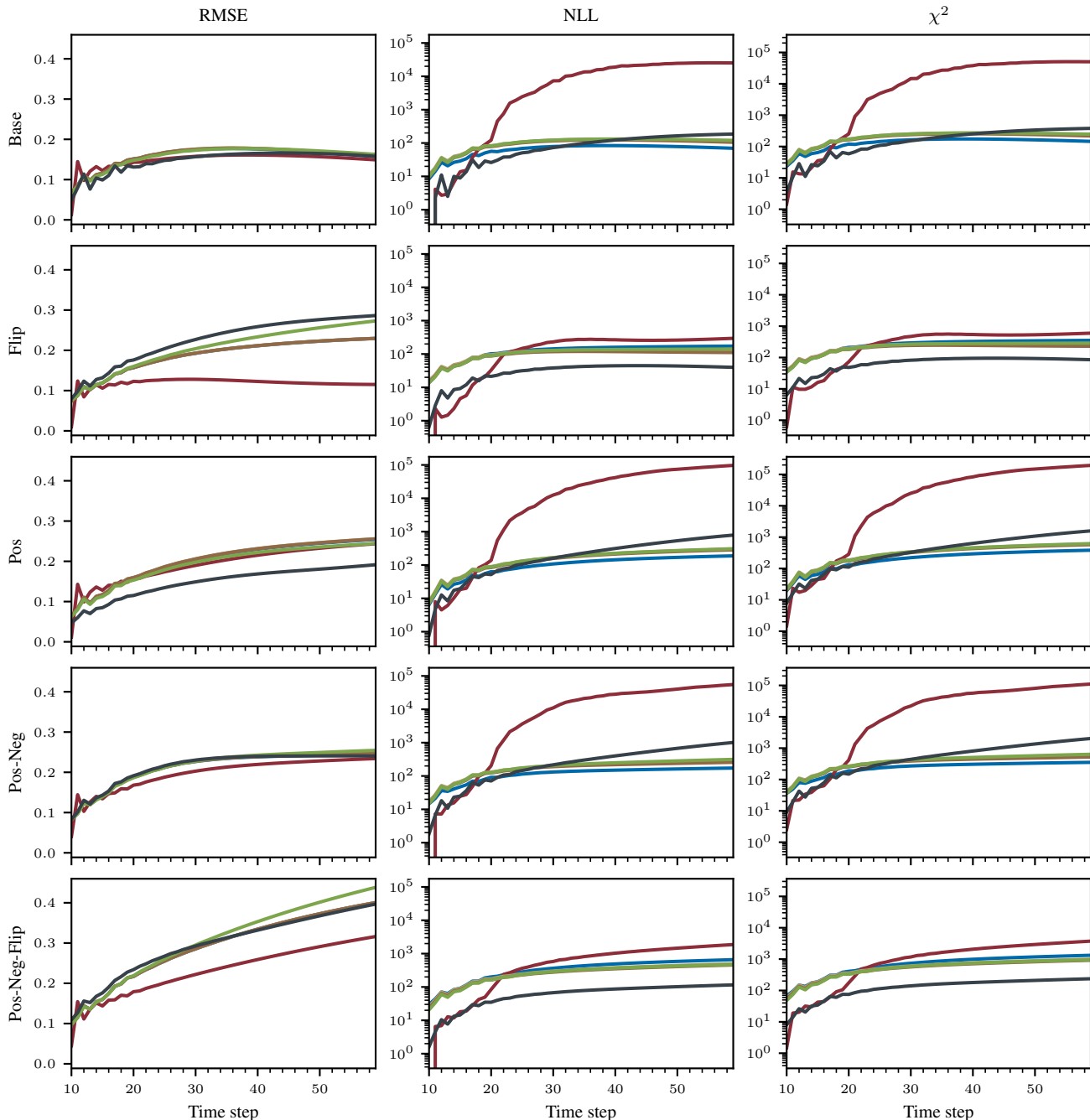

Figure 7: Averaged performance of different UQ methods on an autoregressive rollout of the FNO on 50 trajectories from the Base, Flip, Pos, Pos-Neg, and Pos-Neg-Flip datasets. We compare input perturbations (—), deep ensembles (—), Sample-Iso (—), LUNO-Iso (—), Sample-LA (—), LUNO-LA (—).

