# OpenReview forum: "Linearization Turns Neural Operators into Function-Valued Gaussian Processes"
_ICML.cc/2025/Conference — ICML 2025 spotlightposter_

### Official Review · Reviewer_wc7q · 2025-03-13

**Overall Recommendation:** 4

**Summary:**

The authors propose a new approach to approximate stochastic neural networks with Gaussian weights into Gaussian processes (GPs). The approach is based on performing a linearization around the mean of the weights to obtain a GP approximation of the network. The effectiveness of the framework is shown in some examples where neural networks are used to learn PDEs.

**Claims And Evidence:**

I found the claims well supported by mathematical evidence.

**Essential References Not Discussed:**

The references I mentioned above are not referred to in the paper, and it would be good to include those. Furthermore, a reference that should be discussed  in the main text, and not only in the Supplementary, is [Khan, Mohammad Emtiyaz E., et al. "Approximate inference turns deep networks into Gaussian processes." Advances in Neural Information Processing Systems 32 (2019], where the authors also seem to rely on similar techniques.

**Experimental Designs Or Analyses:**

Experimental analysis is sound. However, as mentioned above, it would be good to extend it a bit more.

**Methods And Evaluation Criteria:**

- An aspect that could be improved is the empirical evaluation. In fact, as, by definition, a linearization is a local approximation, it would be important to have experiments with different weight distributions (with difference variances) to see how the accuracy of the approximation changes with increasing the uncertainty of the distribution. Another interesting experiment would be to compare the approximation obtained with the proposed method with that of the limiting GP obtained by relying on the central theory limit [Lee, Jaehoon, et al. "Deep Neural Networks as Gaussian Processes." International Conference on Learning Representations. 2018.], which is often used to study neural network posteriors [Cardelli, Luca, et al. "Robustness guarantees for Bayesian inference with Gaussian processes." Proceedings of the AAAI conference on artificial intelligence. Vol. 33. No. 01. 2019].

- Minor point: in the future, It would also be interesting to consider non-diagonal distributions on the weights.

**Other Comments Or Suggestions:**

No other comments.

**Other Strengths And Weaknesses:**

I believe that the key strength of the paper is to offer a new way to study the uncertainty of neural networks. Of course, the methods come with the weaknesses described above, but overall, I found the paper to be an interesting contribution.

**Questions For Authors:**

My questions have been already formulated above and in summary, the more important one, apart from improving the related works and the discussion of existing methods, is:

- Can you please extend the experiments as suggested above?

**Relation To Broader Scientific Literature:**

The paper extends recent works that approximate a neural network with a Gaussian process. It does so by defining a Gaussian distribution on a functional space. This allows for the evaluation of uncertainty not only in a finite set of input points, such as in [Adams, Steven, et al. "Finite Neural Networks as Mixtures of Gaussian Processes: From Provable Error Bounds to Prior Selection." arXiv preprint arXiv:2407.18707 (2024).], and not only in the limit of infinitely width size, such as for the papers mentioned above. Of course, the approximation comes with no error bound, which I believe would be a very interesting and promising direction.

**Theoretical Claims:**

Theoretical claims appear sound to me.

---

> ### Author Rebuttal · Authors · 2025-04-01
>
> Thank you for your constructive feedback and for highlighting several interesting directions for further exploration.
> Below are our responses to your main points:
>
> ---
>
> **Comparison with infinite-width GP approaches:**
> We agree that investigating how infinite-width GP methods could be adapted to the infinite-dimensional operator learning setting is an intriguing theoretical direction.
> However, our primary goal is to develop a practical, yet theoretically sound, post-hoc method that could be applied to practical (i.e., non-asymptotic) architectures.
> To our knowledge, infinite-limit GP formulations typically involve retraining networks as the hidden dimension grows, which seems somewhat orthogonal to our focus on uncertainty quantification that leaves the architecture untouched.
> If there is a specific experiment you have in mind, we would be happy to discuss it further during the discussion period.
>
> ---
>
> **Varying weight distributions:**
> We appreciate the suggestion to explore different variances.
> In practice, the choice of weight-space uncertainty might often depend on the downstream application an end-user has in mind.
> For instance, with a Laplace approximation and our low-rank-plus-diagonal covariance structure, the approximation defaults to the prior variance far away from the training data.
> The diagonal portion of our covariance is an isotropic Gaussian prior, so larger variances tend to dilute the impact of the low-rank factors (which encode data-informed structure).
> We tune the prior variance with respect to a validation set, but we agree that experimenting with more informative priors (e.g. non-isotropic or even non-diagonal) could be a valuable direction for future research, also in general for Bayesian deep learning.
>
> ---
>
> **Non-diagonal weight distributions:**
> As mentioned above, we use a low-rank-plus-diagonal structure (which leads to highly non-diagonal weight covariances in practice), but we acknowledge that a more general covariance could capture richer parameter correlations.
> In our experiments, increasing the rank of the covariance did not yield substantial improvements, although we suspect that combining a higher-rank structure with more informative priors may prove beneficial, both of which are supported by our framework.
>
> ---
>
> **Relevant references and related work:**
> Thank you for pointing out additional references.
> We agree that Khan et al. (2019) is an important relevant prior work on Laplace approximations.
> We will move it to the main text in the revised version and also add Adams et al. (2024) as an interesting tangential work (where a GMM is constructed layerwise using their activations) in the related work section.
>
> ---
>
> **Lack of error bounds and robustness guarantees for Bayesian inference:**
> We agree that deriving formal error bounds is a challenging yet important open problem in the entire field of Bayesian deep learning, valuable especially for applications such as operator learning.
> To our knowledge, the field does not currently offer rigorous, general bounds for weight space posteriors, which seems to be important prerequisite for establishing corresponding bounds for our setting.
> In the revised version, we will consider acknowledging this limitation and highlighting the need for theoretical guarantees on the quality of uncertainty estimates.
> While the robustness guarantees in Cardelli et al. (2019) are indeed an interesting starting point, it is unclear how to verify the assumptions of their work for the neural tangent kernels used in this work.
> Specifically, the computation of the supremum in point 3 of the "Constant Computation" section seems intractable or at least highly nontrivial in our case.
> Moreover, the paper seems to assume that the input space is (a subset of) $\mathbb{R}^d$ whereas, in our setting, the input space is an infinite-dimensional function space.
> Nonetheless, we believe that the framework of function-valued Gaussian processes provides a uniquely viable starting point for evaluating and interpreting predictive uncertainties in Bayesian deep learning.
>
> ---
>
> We hope these clarifications address your questions.
> We look forward to continuing the discussion and potentially design an additional experiment with your support that can further refine our work.

---

> > ### Comment · Reviewer_wc7q · 2025-04-01
> >
> > I thank the authors for their replies. I am satisfied with their replies. Consequently, I confirm my positive score.

---

### Official Review · Reviewer_P4r1 · 2025-03-13

**Overall Recommendation:** 4

**Summary:**

The paper introduces LUNO, a framework for uncertainty quantification in neural operators using function-valued Gaussian processes. By leveraging linearization, the method propagates Gaussian weight-space uncertainty to the operator’s predictions, effectively converting trained neural operators into Gaussian random processes. LUNO provides a _post-hoc_, scalable, and resolution-agnostic Bayesian uncertainty estimation approach, evaluated primarily on Fourier neural operators (FNOs) for solving PDEs.

_(for any missing input on any of the fields, please refer to the **Strengths and Weaknesses** or the **Other comments or suggestions** sections)_

**Claims And Evidence:**

- Neural operators lack inherent uncertainty quantification, which limits their reliability in high-stakes applications.
  - *Evidence:* Prior work has focused on deterministic operator learning without probabilistic guarantees.
- Model linearization enables efficient uncertainty propagation without retraining.
  - *Evidence:* Theoretical derivations show that linearization allows weight-space uncertainty to be pushed forward to the output function space.
- The resulting function-valued Gaussian process belief provides structured uncertainty estimates.
  - *Evidence:* The paper introduces a rigorous connection between function-valued Gaussian processes and Bayesian deep learning techniques.

**Essential References Not Discussed:**

Although not exactly my field of expertise, the authors seem to cover extensively the literature relevant for the paper. As a mere suggestion, maybe it is worth discussing Wasserstein Gaussian processes as an alternative probabilistic operator learning approach, as well as maybe other works related to implicit stochastic processes. Moreover, related to the post-hoc approaches, a comment on the effects of the linearization approximation on the uncertainty estimates could be insightful, especially in comparison to other methods (e.g. [1]) that do not alter the predictive mean of the original model.

[1] Ortega, L. A. et al. (2024, July). Variational Linearized Laplace Approximation for Bayesian Deep Learning. In International Conference on Machine Learning (pp. 38815-38836). PMLR.

**Experimental Designs Or Analyses:**

- Evaluates low-data regimes (small training sets) and out-of-distribution (OOD) generalization.
- Compares LUNO to:
  - Sample-based approaches (Monte Carlo sampling of weight posteriors).
  - Input perturbations (simulating uncertainty via randomized inputs).
  - Deep ensembles (training multiple independent models).
- Experiments on Burgers' equation and Advection-Diffusion PDEs, analyzing predictive uncertainty under domain shifts.

**Methods And Evaluation Criteria:**

The framework linearizes the trained neural operator around the mean of a Gaussian weight belief. This is interpreted as a probabilistic generalization of currying, leading to a function-valued Gaussian process. The authors evaluate LUNO on Fourier neural operators (FNOs) applied to PDE problems with multiple metrics, including RMSE, NLL and $\chi^2$ statistics.

**Other Comments Or Suggestions:**

- A lot of the text in the first 3+ pages of the article is devoted to mostly explaining basic concepts and results from previous work. Given the amount of work relegated to the supplementary material, it might be beneficial to streamline these initial sections to focus more on the novel contributions and include more details in the main text. Along the same lines, certain discussions like 5.2 seem more appropriate for the supplementary material.
- The wording of Lemma 4.2 could be improved for clarity.
- Since this work explores the connections between gradient descent and Bayesian inference, it may also be interesting to tackle the "main research question" more from the Bayesian perspective. In particular, it would be interesting to see if there is any complete Bayesian approach to this problem, where the complete loss function is derived from a probabilistic formulation of the problem.
- The authors should extend the experimental part to apply their proposed approach to more complex datasets beyond toy regression tasks.
- Explore whether normalizing flows or implicit stochastic processes (or maybe just implicit distributions) could further enhance flexibility beyond Gaussian assumptions.
- Investigate how the method performs under stochastic gradient descent (SGD) instead of full-batch gradient descent.

**Other Strengths And Weaknesses:**

### Strengths:
- Post-hoc application avoids retraining, making it computationally efficient. This is a specially interesting feature for large-scale applications.
- The presented approach is strongly theoretically grounded, with rigorous uncertainty propagation.
- The method seems to be scalable, working well with high-dimensional data and large models.
- While a bit dense at times, the paper is well written and structured, making it more accessible.

### Weaknesses:
- The method assumes Gaussian weight uncertainty. This hypothesis could limit the model's applicability to highly non-Gaussian posteriors.
- Linearization may introduce approximation errors, particularly for highly non-linear operators. This is a key point that should be adressed further, since one of the main motivations of post-hoc approaches is to avoid retraining and keeping the original model's performance.
- Evaluation focuses on PDE benchmarks, broader validation on other function learning tasks would broaden the method's applicability.

**Questions For Authors:**

1. How does this approach perform on classification tasks where aleatoric noise is often modeled differently?
2. Could the method be extended to convolutional or transformer-based architectures?
3. Could normalizing flows be used in this context to extend the method to more complex uncertainty representations?
4. Is there any approach which would allow for a more efficient storage of $\theta_0$ so that the shifted network method could be more practical in memory-constrained settings? Maybe some sort of approximation or compression technique?

**Relation To Broader Scientific Literature:**

The paper builds on existing work in neural operators and Fourier neural operators (FNOs), connecting with research on Bayesian deep learning techniques such as Laplace approximation, variational inference, and SWAG. It relates to operator learning in function spaces, particularly focusing on function-valued Gaussian processes.

**Theoretical Claims:**

Claims for the submission:
- LUNO constructs function-valued Gaussian processes from neural operators by treating them as infinite-dimensional stochastic processes.
- Probabilistic currying is introduced as a key concept, establishing equivalence with multi-output Gaussian processes.
- Gaussian weight-space uncertainty can be efficiently propagated via linearization, providing a computationally tractable Bayesian formulation.

---

> ### Author Rebuttal · Authors · 2025-04-01
>
> Thank you for your detailed and positive assessment of our work!
>
> ---
>
> **On the assumption of Gaussian weight uncertainty:**
> This is an important point.
> Our theoretical framework (see Step 3 of Section 3.2, and Appendix A, particularly Corollary A.14 and Theorem 3.2, and Section A.4) indeed apply to more general (non-Gaussian) weight-space distributions.
> For instance, the predictive mean and covariance expressions remain valid under arbitrary (non-Gaussian) weight-space beliefs, as long as the respective moments of the weight-space belief exist.
> We focus on Gaussian distributions because they yield a (closed-form) Gaussian process over the output space, enabling well-studied analytic tools (e.g., conditioning, and Bayesian experimental design).
> However, mixture models or other non-Gaussian distributions are an exciting direction for empirical evaluation in future work.
> We will clarify this more explicitly in the revised text.
>
> ---
>
> **Effect of linearization:**
> We want to emphasize that we only linearize the neural operator in the weights and not with respect to its input.
> The linearized neural operator $F^\text{lin}(u, w)$ is linear in $w$, but still highly nonlinear in $u$.
> Furthermore, exactly as in Ortega et al. (2024), the predictions of the original model are not altered, only extended by a covariance which depends on the chosen weight-space uncertainty.
>
> ---
>
> **Applicability beyond Fourier neural operators:**
> While our experiments focus on Fourier Neural Operators (FNOs), the theoretical framework applies in principle to any neural operator, including transformer-based architectures.
> We chose FNOs because of their popularity and the particularly efficient lazy representation they enable for the function-valued posterior process.
> Our implementation exploits the structure of the inverse Fourier transform for computational efficiency, which should in principle extend to other signal transforms such as Spherical FNOs.
>
> ---
>
> **On classification tasks:**
> This is an interesting point.
> Neural operators can naturally be applied to pointwise classification tasks such as semantic segmentation in computer vision.
> It is possible to generalize our methodology to classification tasks, in which the output GP is transformed through a link function, yielding a generalized linear model.
> Computing the pushforward distribution of the output GP through the link function is an active field of research and introduces additional approximations.
> However, a central motivation for this work is the empirical and theoretical study of uncertainty quantification for deep-learning based emulators for scientific simulations, which virtually always focuses on the high-dimensional regression setting.
>
> ---
>
> **Other methods for weight space uncertainty:**
> Thank you for your suggestion.
> Our theoretical framework allows for arbitrary (non-Gaussian) weight-space beliefs, so one could in principle use a normalizing flow to represent the weight-space uncertainty.
> We mostly focus on the Laplace approximation in our experimental analyses because it is cheap to compute and applicable to pretrained models, which makes it particularly promising for large models like neural operators.
> A strength of our framework is that the choice of weight-space uncertainty structure is kept flexible.
> We appreciate the reviewer's suggestion and agree that exploring richer weight-space beliefs is an interesting avenue for future research.
>
> ---
>
> **Compression of the shifted network parameters:**
> In most practical cases, the linearization point $w_0$ (see Appendix A.4, Step 2), is chosen to be equal to the mean $\mu$ of the weight-space belief (e.g., this is set to the weights $w^\star$ of the trained network in case of a Laplace approximation).
> We did formulate Appendix A.4 under an arbitrary linearization point to allow for more flexibility in the theoretical framework, but did not end up using this flexibility in practice.
>
> ---
>
> We hope these clarifications address your questions.
> Thank you again for your time and constructive feedback.
> We also thank you for the additional comments and suggestions regarding streamlining the first three pages and clarifying a few passages.
> We will include these corrections and references.

---

### Official Review · Reviewer_49mR · 2025-03-14

**Overall Recommendation:** 4

**Summary:**

This paper introduces LUNO, a linearization approach for turning a nonlinear neural operator into a Gaussian random operator, thereby providing uncertainty estimates for operator learning. This is important in areas such as safety-critical prediction and out-of-distribution scenarios. The method is compared against baselines, particularly deep ensembles, in a low-data regime and an OOD setting, showing superior and/or more principled uncertainty quantification.

**Claims And Evidence:**

The authors claim to construct a function-valued Gaussian process for uncertainty quantification, linking it to probabilistic currying; this is explained convincingly in the paper.

**Essential References Not Discussed:**

I have not identified any missing references, either in Gaussian processes in a neural operator setting or in linearized Laplace approximation.

**Experimental Designs Or Analyses:**

Yes, I checked all which is included in the main text. See above comments about other neural operator architectures.

Additionally, in line 328 left column, the authors are not clear about what ‘sample-based’ approaches are, so they should include a brief description in the main text. In the appendix, it seems like the authors sample from the distribution of weights and pushforward to the output. But this does not readily give a Gaussian process, unlike what the authors claim afterwards. This could be made clearer.

**Methods And Evaluation Criteria:**

The proposed methods address a low-data regime and an OOD setting. These support the authors' claim that the proposed methods contribute to uncertainty quantification in such domains.

On the other hand, the authors only evaluated their framework on FNO. Since their approach depends on linearization (in terms of network weights), the accuracy of the linear approximation could depend on the network architecture. Therefore, it makes sense to include an experiment evaluating other types of neural operators as well.

**Other Comments Or Suggestions:**

Line 120 left column: ‘This justifies interpreting random processes as probability measures….’, I found this line not so clear - a probability measure is a set function which maps to [0,1] which the random processes spoken of are not. Perhaps better to replace as ‘random variables’.

Line 131 left column: ‘A d’-output Gaussian process is a random process ….for all n\in \mathbb N and a_1,…,a_n \in \mathbb A.’ This reads a bit weird, clearer to say ‘A d’-output Gaussian process f on \mathbb A \times \Omega’ which allows you to refer to the same f later on in the sentence.

Line 134 right column: The F’s in the tuple should be bolded, it is also a bit unclear without further explanations how a function could be seen as jointly Gaussian. Worth including an explanation or point to the appendix somewhere.

Line 244 left column: `f(a, \mu)(x)` is a typo, this should be `f((a,x), \mu)`.

Line 251 left column: upright f is really just `f^{lin}_\mu((a,x), \cdot)`, so it’s a redefinition to simplify notation. Making it clear would improve the reading experience.

Figure 3 needs to be fixed, as there is no longer 3 rows as referred to in the caption. There are also two places in the paper which referred to panel 8, which is the null space projection in Figure 3 - this needs to be fixed too.

**Other Strengths And Weaknesses:**

This paper is very well written and clear, and the experiments are thoughtful. It draws a link elegantly with the concept of currying in functional programming, connecting the two fields. The linearization approach is developed well and rigorously, and although i have not checked the proofs, the claims are sensible and consistent with existing intuitions about Gaussian processes.

**Questions For Authors:**

None.

**Relation To Broader Scientific Literature:**

The paper provides an additional tool in the so-far ill-equipped toolkit for uncertainty quantification for operator learning. As discussed by the authors, current methods do not extend to Fourier neural operators.

The method proposed builds on existing work on estimating the posterior distribution of neural network weights.

**Theoretical Claims:**

I have not checked the correctness of the proofs.

---

> ### Author Rebuttal · Authors · 2025-04-01
>
> Thank you for your positive evaluation of our paper and for pointing out areas where we can further clarify the presentation.
>
> ---
>
> **Applicability beyond FNO:**
> While our experiments focus on Fourier Neural Operators (FNOs), the theoretical framework applies in principle to any neural operator.
> We chose FNOs because of their popularity and the particularly efficient lazy representation they enable for the function-valued posterior process.
> Here, we achieved significant computational speedups by leveraging analytic properties of the inverse fast Fourier transform.
>
> ---
>
> **Clarification on sample-based approaches:**
> We do not claim that sample-based approaches directly yield Gaussian processes, as noted in Section 5:
>
> > We evaluate linearized predictive uncertainty (LUNO-\*) against sample-based approaches (Sample-\*), which require additional approximations to impose a Gaussian Process structure over the output space.
>
> Our linearization approach, by contrast, yields an analytic function-valued GP formulation over the output space, which we believe is a key advantage in operator learning tasks.
> For evaluation purposes we consider sample-based approaches that amount to positing a moment-matched GP using the Monte Carlo estimates of the mean and covariance function in the output space.
> In our experience, this also makes the calibration of SAMPLE-based approaches computationally more expansive.
> We apologize for the vague formulation and will revise the text to make this distinction clearer.
>
> ---
>
> **Typos and notational clarifications:**
> We appreciate your thoroughness in pointing out typos, notational improvements, and figure label inconsistencies. This helps a lot and we will make the necessary corrections in the revised manuscript.
>
> ---
>
> Thank you again for your constructive feedback. Should anything remain unclear, we are happy to clarify during the discussion period.

---

### Official Review · Reviewer_N2Br · 2025-03-16

**Overall Recommendation:** 4

**Summary:**

The paper proposes a novel framework for approximate Bayesian uncertainty quantification in trained neural operators. The approach relies on model linearisation and pishes weight-space uncertainty to neural operators' predictions. This allows the application of Bayesian deep learning methods, such as linearised Laplace approximation, to neural operators.

**Claims And Evidence:**

Claims:
- A novel framework (LUNO) which provides linearized predictive uncertainty in neural operators;
- Interpretation of LUNO as a probabilistic generalization of the concept of *currying* in functional programming;
- Compatibility of LUNO with established methods for quantifying weight-space uncertainty in deep neural networks, including the Laplace approximation.
- LUNO scales to large models and datasets and, like neural operators, is inherently resolutionagnostic.

Evidence:
Proofs and case study on Fourier neural operators. The claims are well supported by the evidence.

**Essential References Not Discussed:**

None are specifically missed as far as I can tell.

**Experimental Designs Or Analyses:**

Yes, the experimental design is suitable and experiments provide detailed evaluation of the proposed approach.

**Methods And Evaluation Criteria:**

Yes, the evaluation criteria are appropriate.

**Other Comments Or Suggestions:**

No other comments.

**Other Strengths And Weaknesses:**

Strength:
- The paper is very clearly structured and motivated.
- The experiments are detailed and well-described.
- The limitations are discussed and acknowledged.

Weaknesses:
- No code is available for reproducibility of the experiments as far as I can tell.

**Questions For Authors:**

Are the authors planning to release the code with the paper in the future?

**Relation To Broader Scientific Literature:**

The paper is in the domain of neural operators and generally touches on uncertainty quantification and utilization of Gaussian processes.

**Theoretical Claims:**

I went over proofs until Corollary A.13. I did not spot any issues and could generally follow the flow of the proofs. However, since this is not my main expertise, I could have missed something. Nevertheless, the way the theoretical arguments are structured is natural and consequential.

---

> ### Author Rebuttal · Authors · 2025-04-01
>
> Thank you very much for your positive review of our paper!
> We are glad to hear that you found the theoretical and experimental aspects clear and well-structured.
> Should anything still remain unclear, we are happy to clarify during the discussion period.
>
> ---
>
> **Publication of code:**
> We intend to release the code for our experiments upon acceptance.
> To preserve anonymity during the double-blind review process, we have refrained from sharing it at this stage due to dependencies and references that could potentially reveal our identity.
> That said, we are fully committed to ensuring reproducibility and will provide the complete codebase.
> If access to the code would be helpful during the discussion period, we would be happy to invest additional time to clean and anonymize it appropriately.

---

### Decision · Program_Chairs · 2025-05-01

**Decision:**

Accept (spotlight poster)

**Comment:**

Four knowledgeable reviewers recommend Accept and I recommend Accept. Reviewer P4r1 had some important comments on clarifications and limitations, which I strongly encourage the authors to include in the Camera-Ready version of the manuscript.